# Hidden synaptic differences in a neural circuit underlie differential behavioral susceptibility to a neural injury

**Akira Sakurai\*, Arianna N Tamvacakis, Paul S Katz**

Neuroscience Institute, Georgia State University, Atlanta, United States

**Abstract** Individuals vary in their responses to stroke and trauma, hampering predictions of outcomes. One reason might be that neural circuits contain hidden variability that becomes relevant only when those individuals are challenged by injury. We found that in the mollusc, *Tritonia diomedea*, subtle differences between animals within the neural circuit underlying swimming behavior had no behavioral relevance under normal conditions but caused differential vulnerability of the behavior to a particular brain lesion. The extent of motor impairment correlated with the site of spike initiation in a specific neuron in the neural circuit, which was determined by the strength of an inhibitory synapse onto this neuron. Artificially increasing or decreasing this inhibitory synaptic conductance with dynamic clamp correspondingly altered the extent of motor impairment by the lesion without affecting normal operation. The results suggest that neural circuit differences could serve as hidden phenotypes for predicting the behavioral outcome of neural damage.

\*For correspondence: akira@gsu.edu

**Competing interests:** The authors declare that no competing interests exist.

**Reviewing editor**: Ronald L Calabrese, Emory University, United States

## Introduction

Experimental and theoretical studies have shown that individual animals can exhibit similar behaviors while differing substantially in the properties of the neurons and synapses underlying those behaviors (*Prinz et al., 2004*; *Goaillard et al., 2009*; *Calabrese et al., 2011*; *Norris et al., 2011*; *Roffman et al., 2012*). The consequences of such hidden variability in neural circuits have not been addressed. One possible consequence is that it impacts the behavioral susceptibility of the animal to trauma. It has been noted that individual people differ from one another to such an extent that it can impair the ability to predict outcomes in cases of traumatic brain injury (*Hukkelhoven et al., 2005*; *Lingsma et al., 2010*; *Forsyth and Kirkham, 2012*) or stroke (*Cramer, 2008a*). Such variability can be hidden under normal conditions but cause differential survival of individuals in the face of critical challenges. In this study, we report that differences in synaptic properties, which were of no consequence under ordinary conditions, caused different outcomes when the circuit was challenged with an injury.

With recent advances in detection techniques, there has been a growing awareness that axonal injury in the white matter plays a complex role in disruption of neural networks underlying higher brain functions (*Adams et al., 2000*; *Schiff et al., 2007*; *Kinnunen et al., 2011*; *Squarcina et al., 2012*). However, there are technical difficulties in manipulating specific neural circuit elements and providing precisely controlled lesions in the mammalian brain. In this study, we use a nudibranch mollusc, *Tritonia diomedea*, in which a neural circuit for rhythmic swimming behavior is widely distributed in the brain. The *Tritonia* swim central pattern generator (CPG) consists of three neuronal types: DSI, C2, and VSI (*Figure 1A*), which form a network oscillator circuit that produces the rhythmic bursting activity (*Figure 1B*) underlying production of the rhythmic movements (*Getting, 1981, 1989b*; *Katz, 2007a, 2007b, 2009*). C2 and VSI both send axons through one of the pedal commissures, Pedal Nerve 6 (PdN6), which connects the two pedal ganglia (*Figure 1C*). Previously, we reported that disconnecting this commissure blocks or seriously impairs the swimming behavior and the motor pattern underlying it (*Sakurai and Katz, 2009b*). In this study, we found substantial individual variability in the synaptic

**eLife digest** The outcome of a traumatic brain injury or a stroke can vary considerably from person to person, making it difficult to provide a reliable prognosis for any individual person. If clinicians were able to predict outcomes with better accuracy, patients would benefit from more tailored treatments. However, the sheer complexity of the mammalian brain has hindered attempts to explain why similar damage to the brain can have such different effects on different individuals.

Now Sakurai et al. have used a mollusc model to show that the extensive variation between individuals could be caused by hidden differences in their neural networks. Crucially, this natural variation has no effect on normal behavior; it only becomes obvious when the brain is injured. The experiments were performed on a type of sea slug called *Tritonia diomedea*.

When these sea slugs encounter a predator they respond by swimming away, rhythmically flexing their whole body. This repetitive motion is driven by a specific neural network in which two neurons— called a cerebral 2 (C2) neuron and a ventral swim interneuron—play important roles. Both of these neurons are quite long and they run alongside each other in the brain, with the ventral swim interneuron being activated by signals sent from the C2 neuron at multiple 'synaptic connections' between the two.

Sakurai et al. showed that the strength of the connections between the C2 neuron and the ventral swim interneuron varied substantially between animals. However, despite this variation, the sea slugs still performed the same number of whole-body flexions as they swam.

Sakurai et al. then made a lesion to the brain, which removed about half of the connections between the C2 neuron and the ventral swim interneuron. This meant that the response of the sea slugs to predators depended on the strength of the remaining connections between the two neurons. Sakurai et al. found that the responses of some sea slugs were only mildly impaired, whereas others were severely impaired. This showed that although variations in the strength of the individual connections had no effect on swimming behavior of normal sea slugs, the same variations had a substantial effect when the brain was damaged. Moreover, by creating computer-generated synapses between the C2 neuron and the ventral swim interneuron, Sakurai et al. were able to change the level of impairment.

These findings suggest that the variability in human responses to brain injury could be due to hidden differences at the neuronal level. In everyday life, these differences are unimportant and individuals are able to function in similar ways in spite of subtle differences in their neuronal configurations. However, when the brain is damaged, the differences become more important. This suggests that certain configurations within neuronal networks are more resistant to brain damage than others.

actions of C2 onto VSI, which correlated with variability in the susceptibility of the behavior to disruption following disconnection of PdN6. Such individual variability in neural circuit elements was hidden under normal conditions, but became functionally relevant only when the system was challenged by injury.

## Results

### Individual variability in the impairment of the swimming behavior upon cutting the commissure

The escape swim behavior of *Tritonia* consists of a series of whole body flexions in response to a noxious stimulus (*Getting, 1989b*; *Katz, 2009*). We previously showed that when one of the pedal commissures, PdN6, was severed (*Figure 1C*), the swimming behavior of the animal was impaired in that the number of body flexions per swim episode decreased compared to sham-operated controls (*Sakurai and Katz, 2009b*). With additional data, we further noticed that the extent of the impairment, in terms of the number of body flexions, varied across individuals (*Figure 2*). In this study, we use the term 'impairment' to mean a decrease in the number of body flexions per swim episode or in the number of VSI bursts per swim motor pattern and the term 'susceptibility' for the likelihood of being impaired upon lesion or blockade of a commissure.

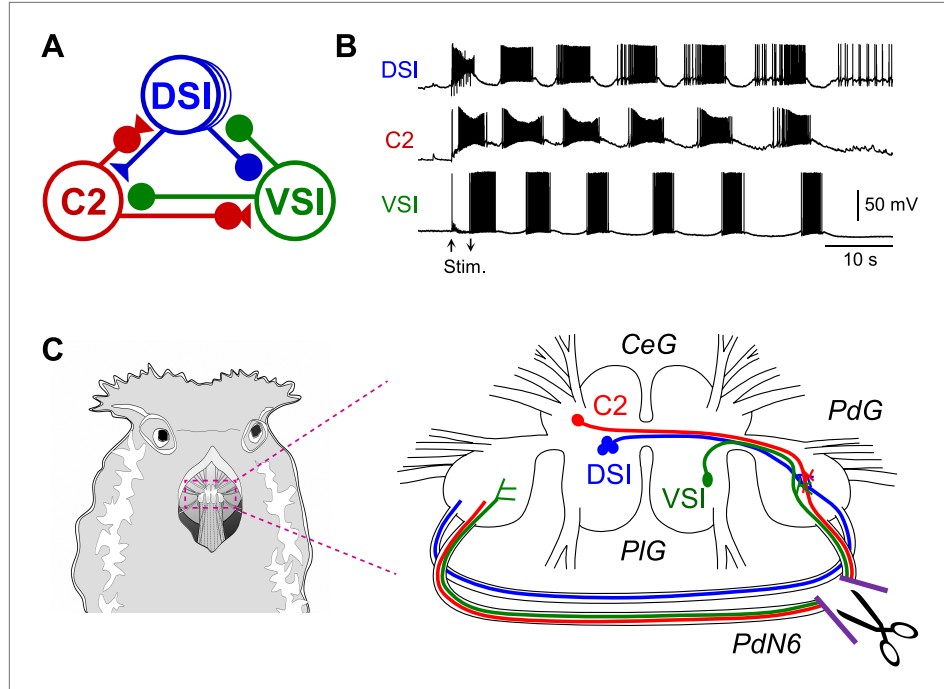

**Figure 1**. The *Tritonia* swim central pattern generator. (**A**) A schematic diagram of the swim central pattern generator (CPG). The CPG consists of three types of interneurons: C2, cerebral cell 2; DSI, dorsal swim interneuron; VSI, ventral swim interneuron. Based on ***Getting et al. (1980)*** and Getting (***1983a***, ***1983b***). All neurons are electrically coupled to contralateral counterparts, which are not represented here. There are three DSIs, but C2 and VSI are individual neurons. Filled triangles represent excitatory synapses and filled circles represent inhibitory synapses. Combinations of triangles and circles are multi-component synapses. (**B**) An example of the swim motor pattern recorded from an isolated brain preparation. Simultaneous intracellular recordings from the three CPG neurons are shown. The bursting pattern was elicited by electrical stimulation of the left body wall nerve, pedal nerve 3 (cf., ***Figure 3A***), using voltage pulses (8 V, 1 ms) at 5 Hz for 3 s. Arrows show onset and offset of the nerve stimulation. (**C**) The *Tritonia* brain and the site where PdN6 was cut in vivo. The body wall above the buccal mass was cut open (left). A schematic drawing shows a dorsal view of the *Tritonia* brain (right) with the locations of the interneurons and their axonal projections. DSI and C2 are located on the dorsal surface of the cerebral ganglion (*CeG*). VSI is located on the ventral side of the pleural ganglion (*PlG*). C2 and VSI project axons through the Pedal commissure (PdN6), which connects the two pedal ganglia (*PdG*) (***Sakurai and Katz, 2009b***). PdN6 was transected near the right pedal ganglion with scissors.

*Figure 2A* shows two examples that illustrate the individual variability in the effect of cutting the commissure on the swimming behavior. In the first example, commissure transection completely disrupted the swimming behavior compared to its sham-operated paired-control, which showed no change (*Figure 2A*, Pair 1). In the second example, the animal with the PdN6 transected had a swim consisting of five flexions compared to six flexions for the sham-operated control (*Figure 2A*, Pair 2).

On average, the surgery decreased the number of flexions in both cut and sham animals (*Figure 2B*). The coefficient of variance (CoV) showed a three-fold increase after the cut (0.21–0.64), but stayed relatively constant for the sham controls (*Figure 2C*). Thus, cutting the commissure revealed a greater variability in the population than could be observed under normal conditions, meaning that some animals were more susceptible to the lesion than others.

## Individual variability in the impairment of the swim motor pattern upon disconnecting the commissure

Individual-to-individual variability was also seen in the extent of impairment in the swim motor pattern recorded from isolated brain preparations (*Figure 3*). Here, action potential propagation in PdN6 was blocked either by physical transection or by local application of TTX to PdN6 (*Figure 3A*, see 'Materials and methods'). *Figure 3B* shows the examples of swim motor patterns, consisting of six cycles of

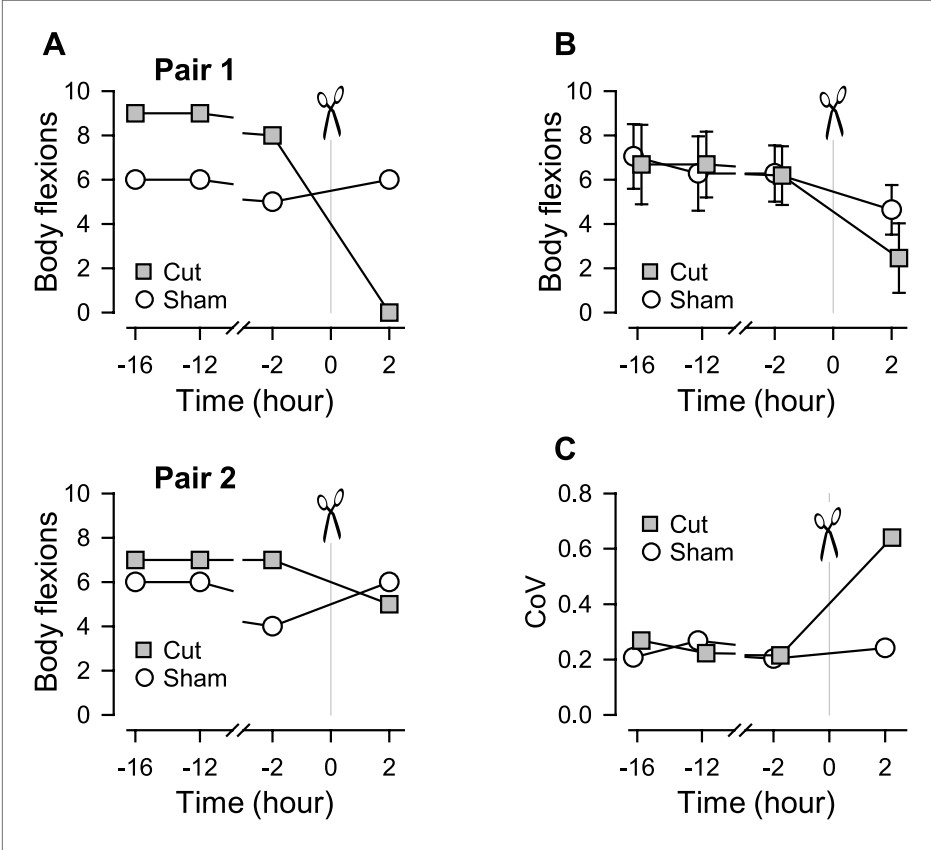

**Figure 2**. Individual variability in the extent of swim impairment by a lesion. (**A**) Nerve-transected animals were blindly paired with sham-operated animals. Two examples (Pair 1 and Pair 2) show different effects on the number of body flexions during the escape swim behavior for animals in response to PdN6 transection (gray squares) compared to sham-operated controls (white circles). In one animal, cutting PdN6 caused a large decrease in the number of body flexions compared to sham (Pair 1), whereas the same lesion caused a small decrease in other experimental preparation (Pair 2). (**B**) Mean number of body flexions during the escape swim behavior for animals with PdN6 transected (gray squares) and sham-operated controls (white circles). The surgery caused a significant decrease in the number of flexions in both cut and sham animals (cut animals, $F_{(3,30)}$ = 21.0, p< 0.001, N = 11; sham animals, $F_{(3,30)}$ = 7.47, p< 0.001, N = 11 by One-way Repeated Measures ANOVA). A two-way repeated measures ANOVA with post-hoc pairwise comparison revealed a significant difference in the number of flexions between cut and sham animals 2 hr after the surgery (p<0.001). Prior to the cut, there was no significant difference between the test and sham-operated animals in the number of flexions (16 hr, p = 0.57; −12 hr, p = 0.52; −2hr, p = 0.89). (**C**) The coefficient of variance (CoV) of the number of body flexions for the transected animals (gray squares) showed a threefold increase after the cut, but only a slight increase in sham-operated animals (white circles). There is a significant difference in variance between the cut group and the sham group after the surgery (by Levene median test, N = 19).

The following source data are available for figure 2:

**Source data 1**.

rhythmic bursts before application of TTX, from two different animals. After blocking PdN6 by replacing saline in the pipette with the saline containing 0.1 mM TTX, the swim motor pattern in Animal 1 was reduced to just two bursts in VSI; whereas in Animal 2, VSI still produced five bursts (*Figure 3B*, right; *Figure 4A*).

In the experiments with local application of TTX, blockade of axonal impulses was confirmed by examining the change in the impulse waveform (*Figure 3B*, overlaid traces in boxes; *Sakurai and Katz, 2009b*). The axonal impulse, recorded *en passant* by a pipette, was triphasic with an apparent positive deflection between two negative deflections. When the action potential was blocked inside

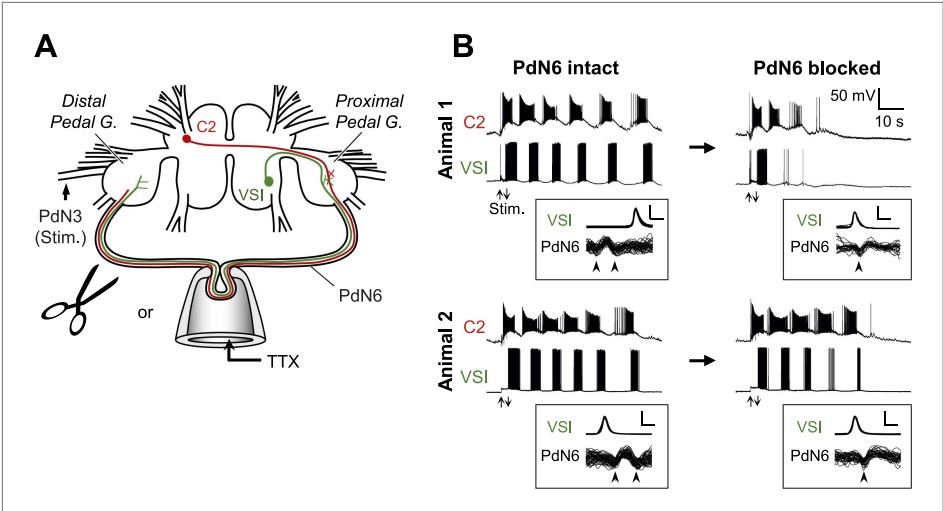

**Figure 3**. Individuals differed in the extent of motor pattern impairment by disconnection of PdN6. (**A**) A schematic drawing of the *Tritonia* brain showing how axonal impulse propagation was blocked in PdN6 either by delivering TTX (1 × 10⁻⁴M) into a suction pipette or by physical transection. The stimulus was delivered to the left pedal nerve 3 (PdN3). The pedal ganglion closer to the VSI cell body was called the proximal pedal ganglion whereas the other pedal ganglion was called the distal pedal ganglion. (**B**) Simultaneous intracellular recordings from C2 and VSI from two representative animals (Animals 1 and 2). Arrows (Stim) indicate the time of PdN3 stimulation. Animal 1 showed a large decrease (from 6 to 2) in the number of VSI bursts after PdN6 was blocked, whereas in Animal 2 the number of VSI bursts was less affected (from 6 to 5). The boxed insets show overlaid traces of VSI spikes recorded from the soma and the corresponding axonal impulses recorded from PdN6 with an *en passant* suction electrode during the swim motor program. The traces were triggered at the peak of the somatic action potential and overlaid. The shapes of the impulses show that the action potentials were blocked (see text for explanation). Calibration: 50 mV, 10 ms.

the pipette, the impulses became biphasic with an initial negative deflection followed by a slower positive deflection. This indicates that the impulses came into the pipette but did not exit. This was also accompanied with the disappearance of the VSI's synaptic action onto neurons in the contralateral pedal ganglion (data not shown, cf., *Sakurai and Katz, 2009b*). Before PdN6 disconnection in Animal 1, axonal impulses in Pd6 appeared earlier than the soma spike, indicating that VSI was producing antidromic action potentials. After blocking PdN6, the axonal impulse appeared after the soma spike, indicating that VSI was now producing orthodromic action potentials. In Animal 2, VSI was already exhibiting orthodromic action potentials even before blocking PdN6, showing individual variability in spike initiation zones (see 'Direction of VSI spike propagation predicted the extent of motor impairment').

On average, disconnection of PdN6 significantly decreased the number of VSI bursts per swim motor pattern episode (*Figure 4B*), as shown previously (*Sakurai and Katz, 2009b*). This decrease was accompanied by an increased CoV from 0.16 to 0.47 (*Figure 4C*). Thus, as with the behavior of the animal, individuals also differed in the extent of impairment of the swim motor pattern in an *ex vivo* preparation after disconnection of PdN6.

## The extent of motor impairment negatively correlated with the C2-evoked depolarization of VSI

C2-evoked excitation of VSI is essential for the production of the swim motor pattern (*Getting, 1983b*; *Calin-Jageman et al., 2007*; *Sakurai and Katz, 2009b*). Stimulating C2 to fire a train of actions potentials (10 Hz for 2 s) induced a burst of action potentials in VSI in 47 out of 53 preparations with PdN6 intact (*Figure 5*). The C2-evoked VSI spikes were mostly antidromic, traveling from the pedal ganglion distal to the VSI soma through PdN6 (*Figure 5A*, N = 39 of 47; cf., *Sakurai and Katz, 2009b*), but their number varied across individuals, ranging from zero to 58 spikes, as seen in the two examples (*Figure 5B*, Animal 3 and Animal 4). The number of bursts in the swim motor pattern, under normal conditions, was not correlated with the number of action potentials evoked by C2 when PdN6 was intact (*Figure 5C*).

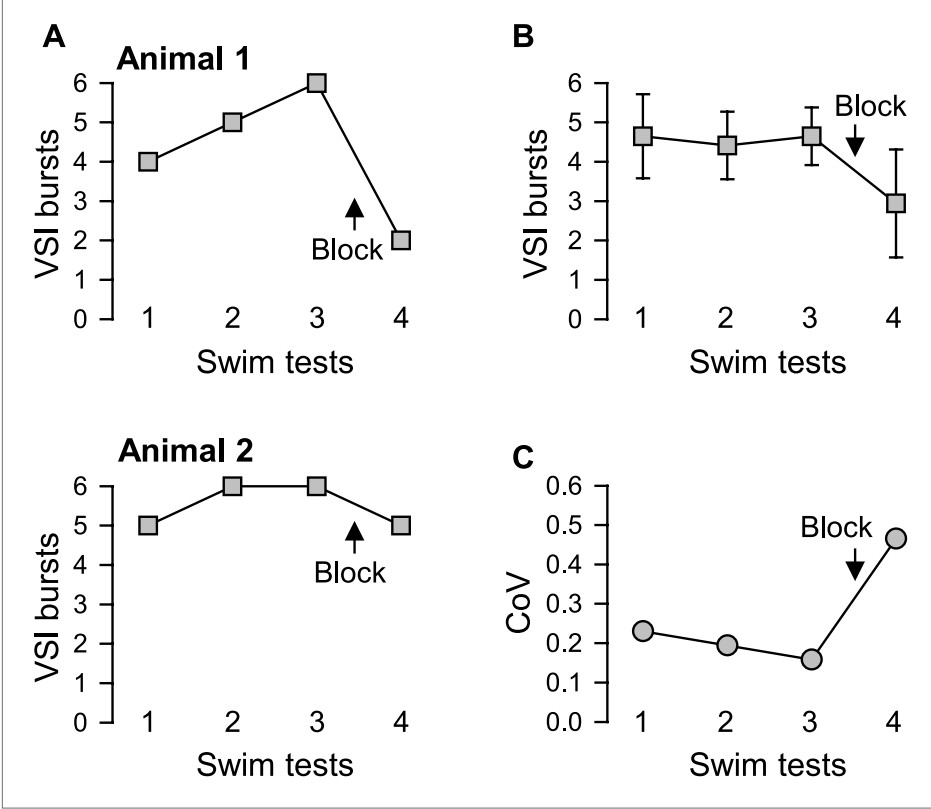

**Figure 4**. Individual variability in the extent of motor pattern impairment by disconnection of PdN6. (**A**) The number of VSI bursts per swim motor pattern episode recorded from Animal 1 was more affected by blocking PdN6 than that from Animal 2 (same individuals as in **Figure 3B**). The swim motor pattern was evoked 3 or 4 times at constant intervals (approximately 10 min), and PdN6 was blocked between the last two swim motor pattern bouts. (**B**) The average number of the VSI bursts decreased significantly after PdN6 block (p<0.001 by one-way repeated measures ANOVA, N = 34). Post-hoc pairwise comparisons (Tukey test) show significant differences of the 4th swim test from all other swim tests. (**C**) The coefficient of variance (CoV) of the number of bursts increased after PdN6 block. There is a significant increase in variance in the number of VSI bursts after PdN6 disconnection (p<0.05 by Levene median test, N = 34).
The following source data are available for figure 4:
**Source data 1**.

Disconnection of PdN6 (**Figure 6A**) decreased the number of VSI bursts per swim episode to different extents in Animals 3 and 4 (**Figure 6Bi,Bii**). Elimination of antidromic spikes by this procedure revealed the C2-evoked membrane potential change evoked in the region of VSI proximal to PdN6 (**Figure 6Biii,Biv**) (cf., **Sakurai and Katz, 2009b**). The synaptic responses of VSI to C2 stimulation were highly variable among individuals. In 47 out of 56 preparations, it was a mix of both depolarization and hyperpolarization (cf., **Figure 6Biii**, Animal 3). In the remaining four preparations, C2 stimulation caused only depolarization with multiple components (cf., **Figure 6Biv**, Animal 4). The response to C2 appeared to contain both monosynaptic and polysynaptic components; during the depolarizing phase, there was a barrage of recruited EPSPs from unknown neurons lasting for about 10 s (cf., **Figure 6— figure supplement 1**).

The amplitude of C2-evoked depolarization in the proximal region of VSI was predictive of the susceptibility of the swim motor pattern to the lesion (**Figure 6C,D**). After PdN6 disconnection, the number of VSI bursts per swim episode showed a significant correlation with the amplitude of the C2-evoked depolarization (**Figure 6C**). The extent of decrease in the number of VSI bursts caused by PdN6 disconnection was also correlated with the amplitude of the C2-evoked depolarization in VSI (**Figure 6D**). In contrast, the amplitude of C2-evoked VSI depolarization showed no

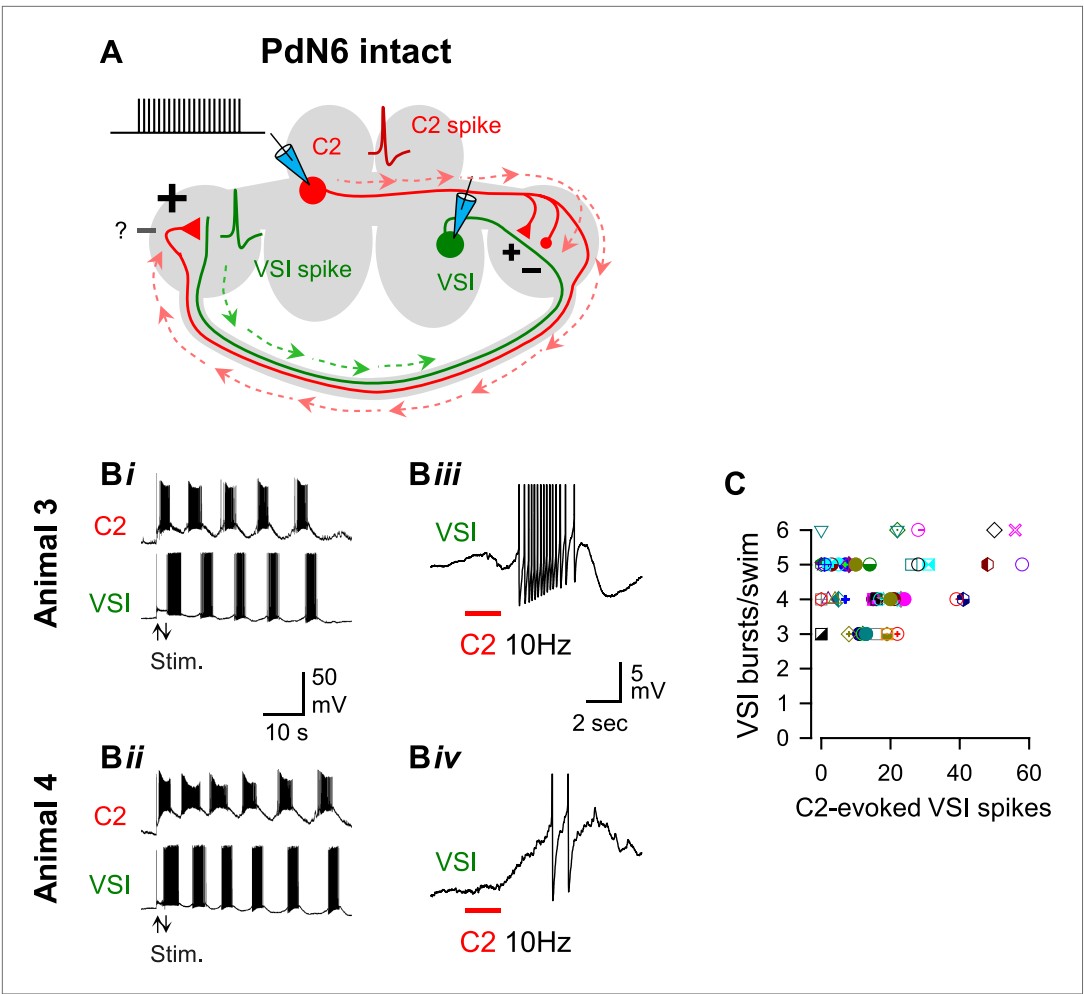

**Figure 5**. The extent of motor impairment showed little or no correlation with the C2-evoked VSI spiking recorded before blocking PdN6. (**A**) A schematic illustration showing the stimulus (C2) and recording (VSI) microelectrodes, the direction of action potential propagation (dashed arrows) in C2 and VSI, and synaptic action (+, excitatory; −, inhibitory) of C2 onto VSI before blocking PdN6. Repetitive square current pulses (10 nA, 20 ms) were injected into the C2 soma to evoke a train of action potentials at a constant frequency (10 Hz). (**B**) Two examples of swim motor patterns (**B*i***, **B*ii***) and the membrane potential responses (**B*iii***, **B*iv***) of VSI to C2 stimulation are shown for two animals (Animals 3 and 4). With PdN6 intact, Animal 3 showed five VSI bursts (**B*i***) and Animal 4 had six VSI bursts (**B*ii***). The effect of C2 stimulation on VSI varied among individuals; causing an intense burst in VSI of Animal 3 (**B*iii***) but only two spikes in Animal 4 (**B*iv***). VSI exhibited antidromic spikes in the majority of preparations (see text) that were presumably caused by the C2 excitatory action in the distal terminal of VSI (***Sakurai and Katz, 2009b***). In **B*iii*** and **B*iv*** action potentials are truncated to show underlying membrane potential. (**C**) No correlation was detected between the number of VSI bursts per swim episode and the number of C2-evoked VSI spikes with PdN6 intact ($R^2 = 0.05$, $p = 0.10$ by linear regression, N = 52). Graph symbols in this and ***Figures 6 and 7*** each represent data from the same individuals.

The following source data are available for figure 5:

**Source data 1**.

correlation with the number of VSI bursts per swim episode with intact PdN6 (p = 3.2; not shown). Thus, the larger the VSI depolarization caused by C2 after PdN6 disconnection, the less impairment there was in the swim motor pattern. This makes intuitive sense; when PdN6 is disconnected, only animals in which C2 still has an excitatory action onto VSI should be capable of swimming because C2 excitation of VSI is necessary for production of the swim motor pattern (***Getting, 1983b***; ***Calin-Jageman et al., 2007***; ***Sakurai and Katz, 2009b***).

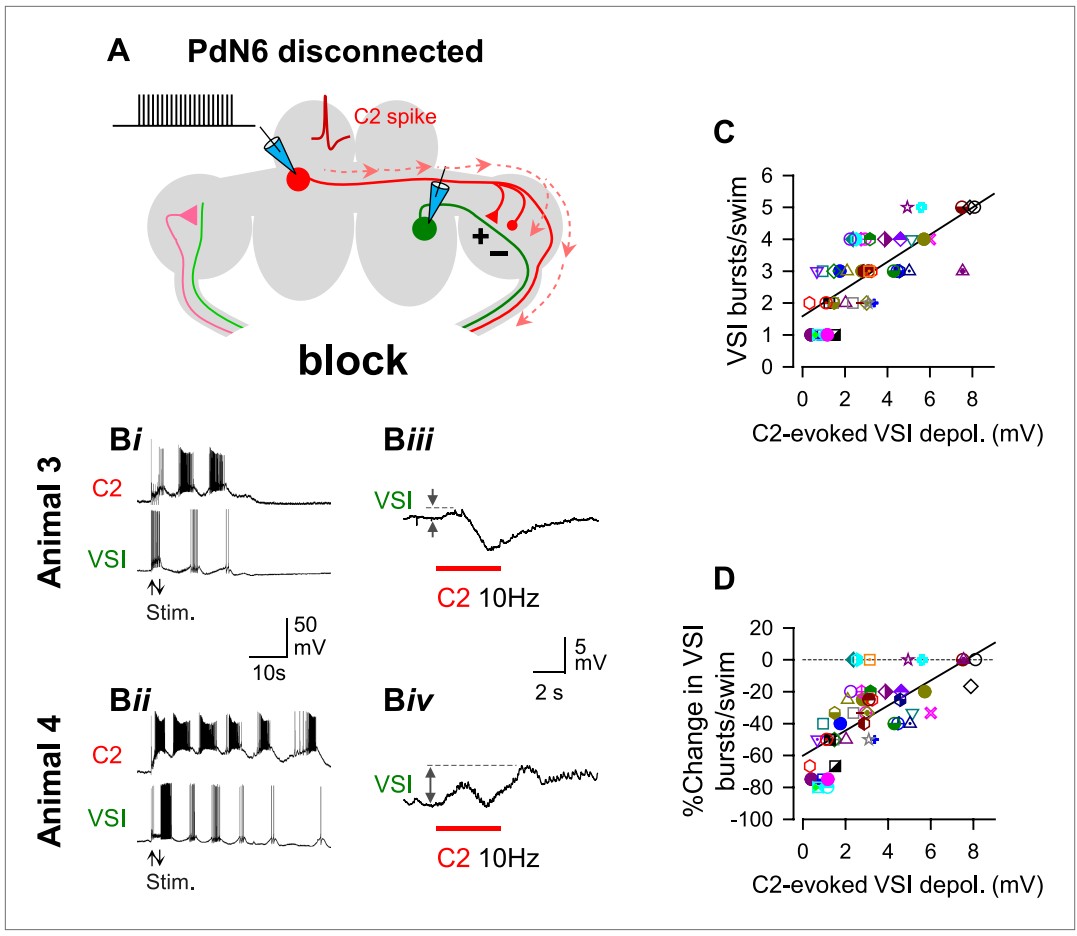

**Figure 6**. The extent of motor impairment showed a strong correlation with C2-evoked VSI depolarization recorded after blocking PdN6. (**A**) A schematic illustration showing the stimulus (C2) and recording (VSI) microelectrodes, the direction of action potential propagation (dashed arrows) in C2, and synaptic action (+, excitatory; -, inhibitory) after blocking PdN6. (**B**) Two examples (Animals 3 and 4) of swim motor patterns (**B***i*, **B***ii*) and the membrane potential responses (**B***iii*, **B***iv*) of VSI to C2 stimulation are shown after blocking PdN6. Animal 3 and 4 are the same animals as in *Figure 5B*. The effects of blocking PdN6 on the swim motor pattern were different: Animal 3 showed 40% reduction (5 to 3) in the number of VSI bursts (**B***i*), whereas Animal 4 showed a 16.7% reduction (6 to 5) (**B***ii*). With PdN6 blocked, C2 stimulation (10 Hz, 4 s) no longer caused VSI to spike in either animal, but instead evoked a complex membrane potential change consisting of both depolarization and hyperpolarization (**B***iii*, **B***iv*). (**C**) After PdN6 disconnection, there was a significant correlation between the number of VSI bursts per swim episode and the amplitude of the C2-evoked VSI depolarization ($R^2 = 0.53$, p< 0.001 by linear regression, N = 50). (**D**) The percent change in the number of VSI bursts caused by PdN6 disconnection showed a significant correlation with the amplitude of the C2-evoked depolarization in VSI (p<0.001 by linear regression, $R^2 = 0.47$, p<0.001, N = 50).

The following source data and figure supplements are available for figure 6:

**Source data 1**.
**Source data 2**.
**Figure supplement 1**. C2 recruited unidentified neurons to excite VSI.

## The extent of motor impairment correlated with the inhibitory component of the C2-to-VSI synapse

The C2-evoked synaptic response of VSI contained both monosynaptic and polysynaptic components (*Figure 6—figure supplement 1*). Even before C2 stimulation, spontaneous excitatory post-synaptic

potentials (EPSPs) from unidentified neurons continuously occurred in VSI. These EPSPs obscured the direct synaptic action of C2 onto VSI. Therefore, we minimized such polysynaptic actions by applying high-divalent cation (Hi-Di) saline, which reduces polysynaptic inputs by raising the firing threshold of neurons (*Sakurai and Katz, 2009a*).

In Hi-Di saline, C2 stimulation evoked a smooth biphasic synaptic potential in VSI, with an initial depolarization and a delayed hyperpolarization phase (*Figure 6—figure supplement 1*, *Figure 7A*). The shapes of these responses were not affected by blocking PdN6, suggesting they were produced in the proximal VSI region, which was electrotonically detectable from the soma recording site. The hyperpolarizations tended to be larger in amplitude and were more variable (2.9 ± 1.3 mV, N = 32, CoV = 0.66) than the depolarizations (0.8 ± 0.36 mV, N = 32, CoV = 0.42); there was a significant difference in variance between the amplitudes of hyperpolarizations and depolarizations (p< 0.05 by Levene median test, N = 32).

The percent decrease in the number of VSI bursts per swim episode showed a significant inverse correlation with the amplitude of the C2-evoked delayed hyperpolarization in VSI in Hi-Di saline (*Figure 7B*). In contrast, we could not detect a correlation between the initial depolarization phase and the number of VSI bursts (p = 0.07, N = 26; not shown). These results indicate that the variability in the susceptibility of the motor pattern to PdN6 disconnection originates at least in part from the difference in the extent to which C2 inhibits VSI; animals in which C2 evoked larger hyperpolarizations in VSI were more vulnerable to having their motor pattern disrupted. Indeed, the amplitude of C2-evoked depolarization in normal saline (*Figure 7—figure supplement 1A*, inset next to y axis) showed no correlation with the amplitude of the depolarizing phase recorded in Hi-Di saline (*Figure 7—figure supplement 1A*), whereas it showed a significant inverse correlation with the amplitude of the hyperpolarization phase (*Figure 7—figure supplement 1B*). Thus, the magnitude of C2-evoked polysynaptic excitation of VSI is more likely determined by the amplitude of C2-evoked hyperpolarization phase, which limits the depolarizing effect of the recruited polysynaptic EPSPs in VSI in normal saline.

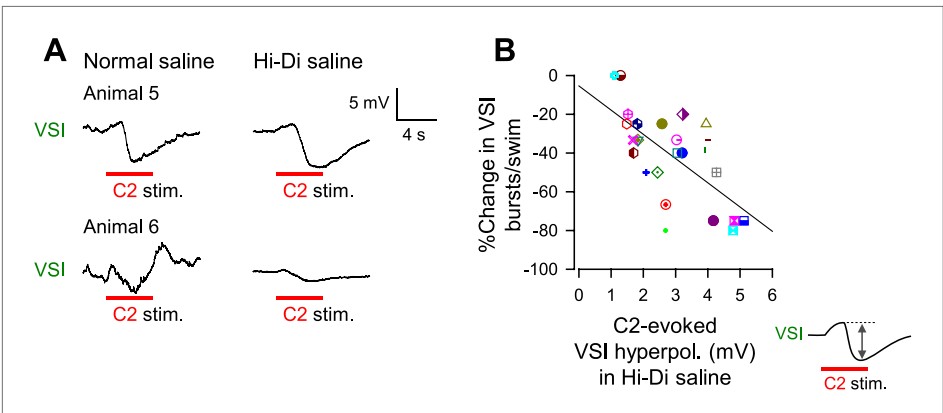

**Figure 7**. The extent of motor impairment correlated with the inhibitory component of C2-to-VSI synapse. (**A**) Two examples (Animals 5 and 6) of VSI membrane potential responses to C2 stimulation recorded with PdN6 disconnected in normal saline (left) and in high divalent cation (Hi-Di) saline (right) to decrease the contribution of polysynaptic inputs. (**B**) The impairment, measured as the percent change in the number of VSI bursts, showed a significant correlation with the amplitude of the hyperpolarization phase ($R^2$ = 0.44, p<0.001 by linear regression, N = 26) of the C2-evoked synaptic potential in VSI.

The following source data and figure supplements are available for figure 7:

**Source data 1**.
**Source data 2**.
**Figure supplement 1**. The magnitude of C2-evoked depolarization in VSI in normal saline correlated with the amplitude of hyperpolarizing phase of C2-to-VSI synaptic potential.

## Direction of VSI spike propagation predicted the extent of motor impairment

The difference in the synaptic action of C2 onto VSI in the proximal pedal ganglion may cause individual differences in the location of the spike initiation zone in the VSI process. We have shown earlier in this study that in some animals the action potential propagation in VSI axon was orthodromic during the swim motor pattern (*Figure 3B*, Animal 2, PdN6 intact). Therefore, we checked the direction of VSI spikes propagation during the swim motor pattern and examined how it correlated with the susceptibility of the motor pattern to PdN6 disconnection.

We previously showed that in most preparations, stimulation of C2-induced VSI action potentials that propagated antidromically through the axon in the PdN6 to the cell body (*Sakurai and Katz, 2009b*). By simultaneously recording VSI spikes from the soma and its axonal impulses from PdN6 (*Figure 8A,B*), we found that the direction of VSI action potential propagation during the swim motor pattern varies among individuals (*Figure 8C*). Out of 69 animals, 17 (24.6%) exhibited only antidromic spike propagation during the swim motor program in which axonal impulses appeared earlier in PdN6 than in the cell body (*Figure 8Ci*). In contrast, five animals (7.2%) showed only orthodromic spikes in which action potentials were generated in or near the pedal ganglion proximal to the VSI soma (*Figure 8Ciii*). In 47 animals (68.1%), the direction switched from orthodromic to antidromic, or *vice versa*, during the swim motor pattern (*Figure 8Cii*). There was a significant difference in the percentage change in the number of VSI bursts per swim episode after disconnecting PdN6 when comparing animals showing only orthodromic VSI spikes with those having only antidromic VSI spikes; animals with only orthodromic VSI spikes were significantly less impaired than those with only antidromic VSI spikes (*Figure 8D*).

VSI tended to show more antidromic spiking later in the swim motor pattern in the majority of animals (*Figure 8—figure supplement 1A,B*). For individual VSI bursts, there are three types of bursts: bursts with all antidromic spikes, bursts with mixed spikes, and bursts with all orthodromic spikes (*Figure 8—figure supplement 1A*). In a majority of animals, VSI exhibited all antidromic spiking after the 3rd burst (*Figure 8—figure supplement 1B*). Upon blocking PdN6, the motor pattern tended to lose the terminal bursts that contained only antidromic spikes (*Figure 8—figure supplement 1C*). Thus, the susceptibility of the motor pattern to lesion was apparently dependent on the location of the primary spike initiation zone in VSI. If spikes originated in the proximal region of VSI, PdN6 disconnection would have less effect on the swim performance. Although we could not provide direct evidence of how the primary spike initiation zone was determined, it likely involves the C2 synaptic action onto VSI.

## Artificial enhancement of synaptic inhibition from C2 to VSI changed the extent of motor impairment without affecting normal function

The results above suggest that the strength of the inhibitory component of the C2-to-VSI synapse does not affect the function of the intact swim circuit under normal conditions, but may determine its susceptibility to the lesion. To test this, we employed the dynamic clamp technique (*Sharp et al., 1992*, *1993a*, *1993b*) to introduce an artificial C2 to VSI synaptic conductance. The time course of the conductance was based on that from previous models of the *Tritonia* swim CPG (*Getting, 1989c*; *Calin-Jageman et al., 2007*). The activation and maximum conductance were adjusted to mimic the synaptic strength observed in that preparation (see 'Materials and methods').

The example traces shown in *Figures 9 and 10* were obtained from the same preparation, which was slightly susceptible to PdN6 disconnection; it exhibited 5 VSI bursts per swim episode with PdN6 intact (*Figure 9Ai*) and 4 bursts after PdN6 was blocked (*Figure 10Ai*). Introducing an artificial inhibitory synaptic conductance in VSI corresponding to the times of C2 spikes caused no change in the number of bursts when PdN6 was intact (*Figure 9Aii,Bi*). When the numbers of bursts recorded with dynamic clamp were plotted against the number of bursts without dynamic clamp, they lined up along the unity slope line (*Figure 9Bii*). Before dynamic clamping, orthodromic VSI action potentials were detected in 50% of the preparations during the swim motor pattern (N = 8 of 16). When the artificial inhibitory synaptic conductance was added to the soma by dynamic clamping, all VSI action potentials became antidromic during the swim motor pattern in each of the 16 preparations. This indicates that enhanced hyperpolarization in the soma suppresses orthodromic spiking in VSI, but the distal VSI terminal is still able to generate antidromic bursts.

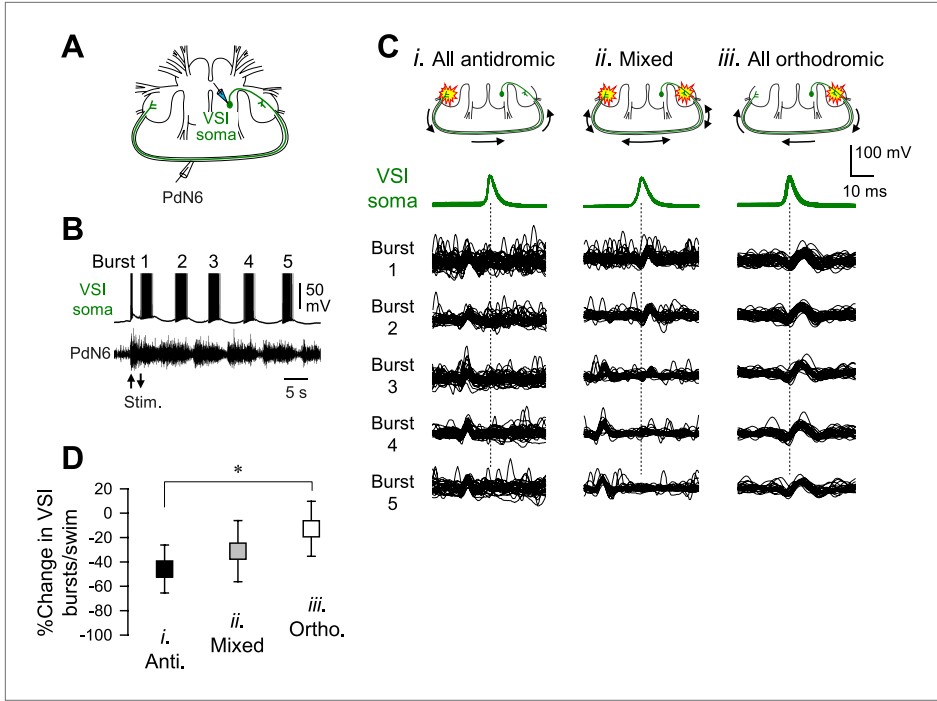

**Figure 8**. The direction of spike propagation in VSI axon was predictive of susceptibility of the swim motor pattern to PdN6 disconnection. (**A**) A schematic diagram showing the recording configuration. VSI action potentials were recorded with an intracellular microelectrode in the soma and an extracellular *en passant* suction electrode on PdN6. To initiate a swim motor pattern, the left PdN3 was stimulated via a suction electrode (see ***Figure 3A***). (**B**) Intracellular activity recorded from VSI and the axonal impulses recorded extracellularly from PdN6 during a swim motor pattern. Arrows indicate the time of PdN3 stimulation to initiate the swim program. Each VSI burst is indicated by a number (1–5). (**C**) Overlaid spike-triggered impulses for each burst recorded from PdN6 in three individuals show variability in the direction of VSI spike propagation (**C***i*, antidromic; **C***ii*, mixed; **C***iii*, orthodromic). Schematic drawings above the traces show the presumptive spike-initiation zones (yellow explosion symbols) and the direction of action potential propagation (arrows) in the VSI axons. In **C***i*, all five bursts in the swim program consisted of antidromic VSI spikes (the nerve impulse appearing earlier than the soma spike), whereas in **C***ii*, VSI spike propagation shifted from orthodromic to antidromic during the course of the swim motor pattern. In **C***iii*, all VSI spikes were evoked near the soma and propagated orthodromically. Traces in **C***ii* were reused from ***Sakurai and Katz (2009b)***. (**D**) The direction of VSI spike propagation in PdN6 was predictive of the extent of impairment after PdN disconnection. The extent of impairment by PdN6 disconnection, shown as the percent change in the number of VSI bursts per swim episode, is plotted in three groups categorized by the direction of VSI spike propagations (black, all VSI bursts were antidromic; gray, mixed; white, all bursts were orthodromic). One-way ANOVA with a post-hoc pairwise comparison (Holm-Sidak method) revealed that individuals exhibiting only orthodromic VSI bursts were significantly less impaired than those with only antidromic VSI bursts as indicated by an asterisk ($F_{(2,66)}$ = 4.64, p = 0.015, N = 5 and 16).

The following source data and figure supplements are available for figure 8:

**Source data 1**.

**Source data 2**.

**Figure supplement 1**. Inter- and intra-individual variation in the direction of VSI spike propagation during the swim motor pattern.

---

We next tested the effects of subtracting the delayed inhibitory component onto the swim motor pattern by setting the inhibitory conductance to a negative value (see 'Materials and methods'). Subtraction of the C2-evoked inhibition of VSI significantly increased the number of VSI bursts per swim episode in 6 out of 9 preparations (***Figure 9A****iii*,**C***i*). The swim motor pattern was lengthened approximately 20% (***Figure 9C****i*, D.C.), causing an upward shift in the plot comparing the number of

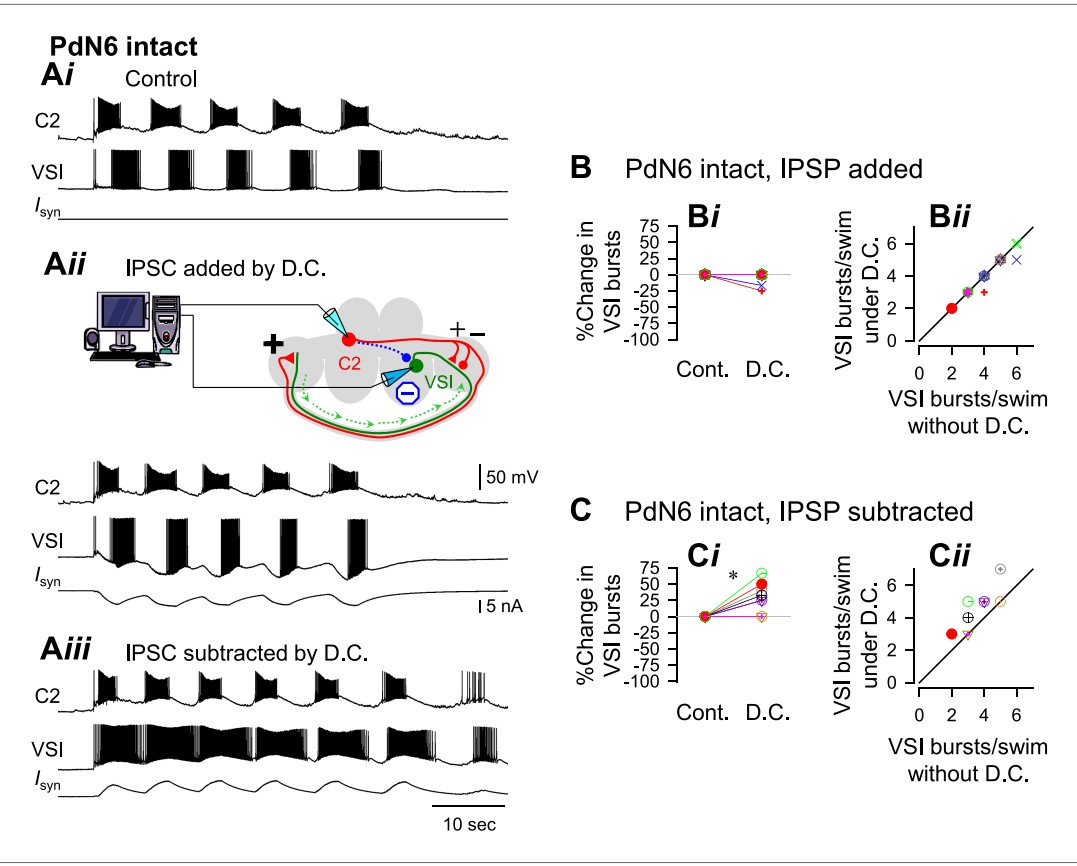

**Figure 9**. An artificial synaptic conductance created a hidden circuit change that caused no motor impairment with PdN6 intact. **A**) Recordings of a five-cycle swim motor pattern with PdN6 intact (**A*i***). Introduction of an artificial synaptic conductance from C2 to VSI using dynamic clamp (D.C.) had no effect on the number of VSI bursts (**A*ii***). The artificial synapse is represented as a dotted blue line with a filled blue circle in the schematic. VSI displayed unnaturally large hyperpolarizations on each burst because the currents were injected at the site of the electrode impalement in the soma instead of occurring in the neuropil. The spikes rode on the hyperpolarizing phase of the burst, indicating that they were antidromic spikes, generated in the distal pedal ganglion (green arrows in schematic). When the inhibitory synaptic conductances were subtracted by applying negative conductances of the same amount as in **A*ii***, the number of bursts increased (**A*iii***). **B**, With PdN6 intact, addition of synaptic inhibition with dynamic clamp (D.C.) did not change the number of VSI bursts compared to control (Cont) (**B*i***, p = 0.16 by paired *t*-test, N = 18). A plot of the number of bursts under dynamic clamp vs the number of bursts without dynamic clamp has a slope close to one (**B*ii***). **C**, With PdN6 intact, subtraction of synaptic inhibition significantly increased the number of VSI bursts 26.6% compared to control. (p = 0.01 by paired *t*-test, N = 9) (**C*i***). All of the preparations either produced the same number of bursts or increased by up to 2 bursts (**C*ii***). Graph symbols in ***Figures 9 and 10*** each represent data from same specimens.

The following source data are available for figure 9:

**Source data 1**.

VSI bursts with and without dynamic clamp (***Figure 9Cii***). Under such conditions, all VSI bursts contained orthodromic action potentials in all 16 preparations. Thus, with PdN6 intact, additional synaptic inhibition from C2 to VSI did not affect the number of bursts in the motor pattern; it acted like a hidden change in the circuit. In contrast, subtraction of the inhibitory conductance extended the motor pattern by causing more orthodromic spiking.

When PdN6 was blocked, addition of the artificial C2 to VSI synaptic inhibition further decreased the number of VSI bursts per swim episode, making the preparation more susceptible to disconnection of PdN6. When the dynamic clamp was turned on, it decreased the number of VSI bursts from four to one (***Figure 10Aii***). The number of C2 bursts was largely unaffected, probably because the artificial

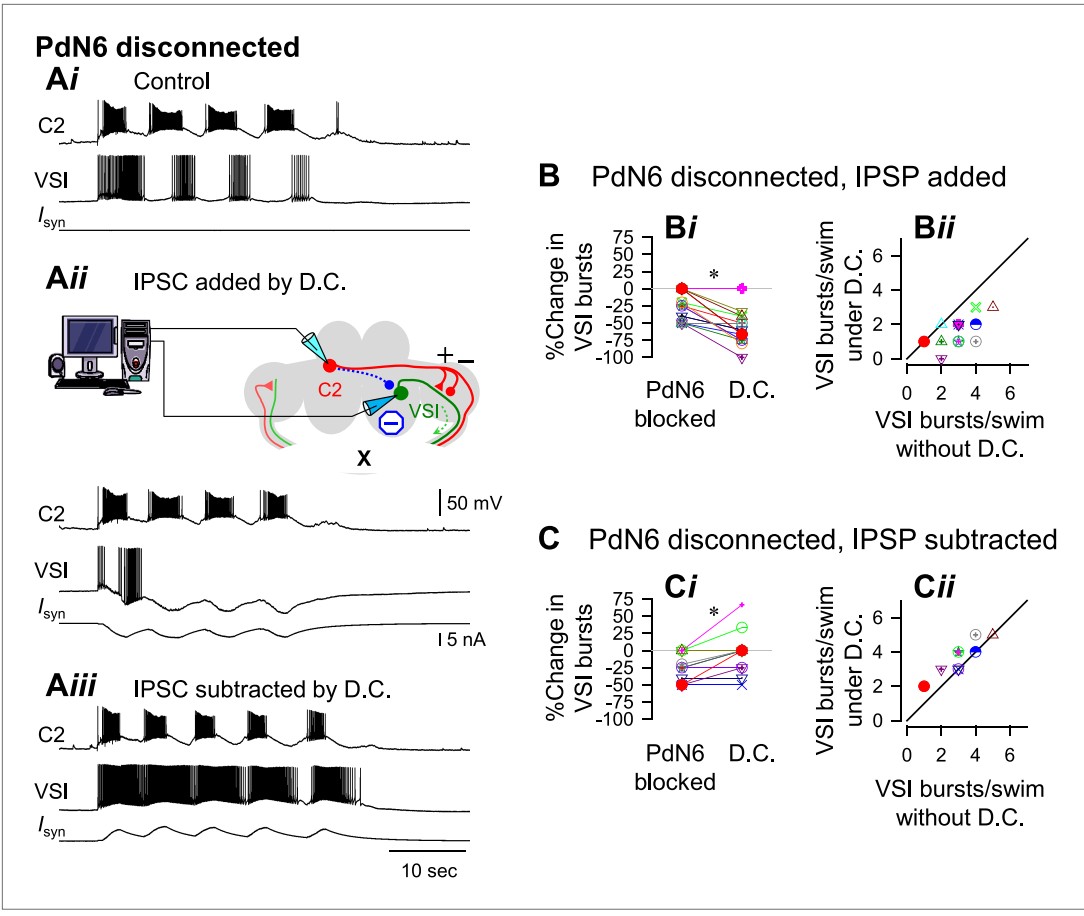

**Figure 10**. With PdN6 disconnected, an artificial synaptic conductance reduced the number of VSI bursts. (**A**) Recordings from the same preparation as **Figure 9**, but with PdN6 disconnected. PdN6 disconnection reduced the number of VSI bursts from five to four bursts per swim episode (**A***i*) (Compare with **Figure 9A***i*). Addition of an artificial inhibitory synaptic conductance using dynamic clamp (D.C) further decreased the number of VSI bursts to one (**A***ii*). Subtraction of the inhibitory synaptic conductance with the dynamic clamp restored the number of VSI bursts to five (**A***iii*). (**B**) With PdN6 blocked, addition of synaptic inhibition with dynamic clamp significantly decreased the number of VSI bursts (**B***i*, PdN6 blocked vs D.C., p<0.0001 by paired t-test, N = 20). PdN6 disconnection decreased the number of VSI bursts by 24.7 ± 20.8% from control. Addition of an artificial synaptic conductance using dynamic clamp decreased the number of VSI bursts further to 57.7 ± 22.0%. Comparison of the number of VSI bursts with dynamic clamp to control shows the points falling below the unity line (**B***ii*). (**C**) With PdN6 blocked, subtracting the synaptic inhibition restored the motor pattern. With dynamic clamp, number of VSI bursts was increased from −21.9 ± 20.7% below control to −3.1 ± 30.0% (**C***i*, p = 0.003 by paired *t*-test, N = 19). For most preparations, the effect of dynamic clamp was to increase the number of VSI bursts (**C***ii*).
The following source data are available for figure 10:
**Source data 1**.

conductance was applied to only one of bilateral VSI pair and C2 is electrically coupled to its contralateral counterpart. On average, without dynamic clamp, blocking PdN6 caused a 25% decrease in the number of VSI bursts (**Figure 10B***i*, PdN6 blocked). When the dynamic clamp was turned on, this became an approximate 60% decrease compared to control (**Figure 10B***i*, D.C.). Thus, after PdN6 disconnection, addition of inhibitory synaptic current significantly decreased the number of bursts. Plotting the numbers of VSI bursts with dynamic clamp against the numbers of bursts without dynamic clamp shows that all but two of the 18 preparations fell below the unity slope line (**Figure 10B***ii*). Thus, artificial enhancement of C2-to-VSI inhibition made the swim motor pattern more susceptible to disruption caused by PdN6 disconnection.

If the extent of motor impairment by PdN6 disconnection were proportional to the extent of C2-evoked synaptic inhibition of VSI, then decreasing the delayed inhibitory component through subtraction (see 'Materials and methods') ought to prevent the motor impairment. Indeed, subtracting the inhibition from C2 to VSI mitigated the effect of disconnecting PdN6. When the dynamic clamp was turned on, the decreased number of VSI bursts was restored from 27.7% below the control swim motor pattern to 3.3% above the control, a significant shift (*Figure 10Ci*). The effect of subtracting the inhibition tended to be more effective in the more vulnerable preparations, which increased the number of bursts by one (*Figure 10Cii*). Thus, subtracting the inhibitory component of the C2 to VSI synapse restored the impaired swim motor pattern after PdN6 disconnection.

## Discussion

Although the escape swimming behavior of *Tritonia* is normally very robust with little variability among individuals, we found that there was an increase in the variability of the behavior after lesion of a central commissure, PdN6, which connects the two pedal ganglia. Specifically, some animals were more susceptible to the lesion than were others, producing fewer body flexions per swim episode. A similar increase in animal-to-animal variability was observed in the motor pattern episodes recorded from isolated brain preparations following the commissure lesion.

In the neural circuit for the escape swimming behavior, individually identified neurons (VSI and C2) play essential roles in the neural circuit underlying the swimming behavior. We found that there is animal-to-animal variation in the strength and the topological distribution of synapses between C2 and VSI. Such variation does not affect the motor pattern under normal conditions, but causes variability in the susceptibility of the motor behavior to lesion of the commissure. To our knowledge this is the first direct analysis of synaptic variation affecting vulnerability of a neuronal circuit to a particular lesion.

It has been known that the magnitude of functional impairment varies among individuals to such an extent that one cannot predict outcomes in cases of traumatic brain injury (*Hukkelhoven et al., 2005*; *Lingsma et al., 2010*; *Forsyth and Kirkham, 2012*) or stroke (*Cramer, 2008a*). Severe loss of brain function is often caused by a complex pattern of diffuse axonal injury in the white matter that are critical nodes for distributed network functions (*Adams et al., 2000*; *Schiff et al., 2007*; *Kinnunen et al., 2011*; *Squarcina et al., 2012*). Stroke can also cause axonal lesions in subcortical white matter (*Bamford et al., 1991*; *Sozmen et al., 2009*; *Blasi et al., 2014*). Disruption of axonal pathways that link nodes in the distributed brain networks would cause motor deficits and cognitive/learning disabilities, which are commonly seen in children with cerebral palsy (*Riddle et al., 2012*). However, case-by-case differences in the extent of lesion are a major problem in assessing the outcome of injury with regard to lesion types and locations (*Saatman et al., 2008*; *Bigler et al., 2013*), which also involve the extent of secondary responses such as inflammation and degradation (*Lenzlinger et al., 2001*; *Woodcock and Morganti-Kossmann, 2013*). Moreover, in mammalian systems, experimental manipulation of neural circuit elements is difficult because of enormous number of neurons with the same or similar functions working as a cluster. Thus, it has been difficult to study how inter-individual differences in neural network properties affect the individual differences in susceptibility to a lesion.

### Variability in susceptibility to injury arises from differences in the inhibitory synaptic component

The variation of susceptibility to a neural lesion appeared to arise from differences in the synaptic action from neuron C2 to VSI in the swim CPG. The difference was hidden under normal conditions. C2-evoked excitation of VSI is thought to be essential for initiating the ventral phase of each swim cycle during the swim motor pattern (*Getting, 1989a*; *Calin-Jageman et al., 2007*). *Getting (1989a)* suggested that C2 excites VSI via direct synaptic action, but we found that the excitation of the proximal VSI process was mainly caused by a bombardment of recruited EPSPs that overrode the direct synaptic action of C2 onto VSI (*Figure 6—figure supplement 1*). However, the polysynaptic recruitment did not appear to play a major role in causing the individual differences in the extent of motor impairment after PdN6 disconnection. Rather, it was the inhibitory component of the direct synaptic action by C2 onto VSI that was more significant in predicting the susceptibility to the lesion (*Figure 7*). In Hi-Di saline, the C2-evoked synaptic potential in the proximal region of VSI can be detected from the cell body as a slow membrane potential change with an initial depolarization and a later hyperpolarization (*Getting, 1989c*; *Calin-Jageman et al., 2007*; *Sakurai and Katz, 2009b*). Stronger inhibition would counteract the depolarizing effect of recruited polysynaptic EPSPs in VSI and hence make the

system more susceptible to loss of the distal spike initiation zone. Synaptic recruitment and concurrent actions of excitatory and inhibitory synapses have been shown to provide flexibility in neural network outputs in both vertebrates and invertebrates (*Berg et al., 2007*; *McLean et al., 2007*, *2008*; *Sasaki et al., 2009*; *Dougherty and Kiehn, 2010*; *Kiehn, 2011*; *Petersen et al., 2014*).

Artificially increasing the extent of inhibition from C2 to VSI in the pedal ganglion, using the dynamic clamp technique, caused the motor pattern to become more susceptible. Importantly, it did not affect the production of the motor pattern with PdN6 intact. This shows that this site of variability is not critical to the animal's behavior under normal conditions, but it becomes critical when challenged by lesion. Thus, this synaptic difference serves as a hidden phenotype that predicts the susceptibility of the neural circuit to disruption.

Subtracting the C2 to VSI inhibition using dynamic clamp partly rescued vulnerable preparations from the effects of the lesion. This further supports the role of this synapse in determining the susceptibility. Subtraction also caused an increase in the number of bursts per swim episode prior to the lesion. This is likely because artificial reduction of the inhibitory synaptic conductance would enhance C2-evoked depolarization through bombardment of recruited EPSPs, which would make the proximal spike initiation zone of VSI more excitable. This enhanced VSI bursting may have a cascading effect in swim motor pattern generation; by spiking more, VSI would decrease the burst duration in C2 and DSI, which may lead toward generating more burst cycles by releasing less serotonin in each burst (*Fickbohm and Katz, 2000*; *Katz et al., 2004*).

## The capability of spike initiation in the proximal region determines the susceptibility

The difference in the synaptic action in the pedal ganglia may affect the direction of spike propagation between them. VSI has two spike initiation zones, one in each pedal ganglion, where C2 synaptic actions take place (*Sakurai and Katz, 2009b*). The proximal and distal spike-initiation zones each are capable of initiating spikes; however, firing of either zone produces spikes in the same axon but with different directions of propagation (*Figure 8*; *Sakurai and Katz, 2009b*). This resembles crayfish central interneurons that integrate segmental mechanosensory inputs (*Hughes and Wiersma, 1960*; *Calabrese and Kennedy, 1974*), locust lobular giant movement detector neurons that generate spikes in both end of the axon (*O'Shea, 1975*), and leech heart interneurons (*Calabrese, 1980*). In leech heart interneurons, a hyperpolarizing current injection near the dominant spike-initiation site revealed the secondary spike initiation site (*Calabrese, 1980*).

It is not unusual for neurons to have multiple spike initiation zones with multiple synaptic sites. Since the early description of the crustacean cardiac ganglion (*Bullock and Terzuolo, 1957*), there have been numerous reports on multiple spike initiation zones in single neurons. Such neurons include crustacean stomatogastric ganglion neurons (*Moulins et al., 1979*; *Meyrand et al., 1992*; *Combes et al., 1993*; *Smarandache and Stein, 2007*; *Thuma et al., 2009*), molluscan giant neurons (*Tauc and Hughes, 1963*; *Zecevic, 1996*; *Antic et al., 2000*), and insects sensory neurons (*Heitler and Goodman, 1978*; *Killian et al., 2000*). In these neurons, individual spike-initiation zones can independently fire, which can also be the target for neuromodulation (*Daur et al., 2009*; *Bucher and Goaillard, 2011*). Local interactions between multiple spike initiation zones are commonly seen in vertebrate cortex (*Llinas et al., 1968*; *Schiller et al., 1997*; *Stuart et al., 1997*; *Larkum and Zhu, 2002*) and olfactory mitral cells (*Andreasen and Lambert, 1998*). Recently, axonal spike back-propagation was shown to play an important role in memory consolidation (*Bukalo et al., 2013*).

In this study, we directly showed that the presence of multiple spike-initiation zones made the circuit more resilient to a lesion by providing a backup spike-initiation zone after loss of the primary spike-initiation zone when the commissure was disconnected. To our knowledge, there have been no studies on whether there is individual variability in controlling multiple spike initiation zones and how an injury to such systems would impair important brain functions. The present results may provide insights of how individual differences in spike initiation could covertly reside in a neural circuit where neurons have multiple spike-initiation zones and how partial damage could therefore differentially impair performance.

## Variation in neural circuits

The findings of hidden variability in well-characterized neural circuits may have profound implications for understanding how neural circuits function. In the mammalian brain, there is a general

acknowledgement of variability. Individual variability in the susceptibility to neural damage may be attributed to substantial variation in the brain function and anatomy (*Cramer, 2008a*). For example, the map of motor cortex is actually the population mean; among individuals there is substantial variation in the relationship between brain function and brain anatomy (*Whitaker and Selnes, 1976*; *Rademacher et al., 1993*; *Van Essen et al., 2012*; *Cramer et al., 2003*; *Cramer, 2008a, 2008b*; *Smith et al., 2014*; *Walhovd et al., 2014*). Endophenotypes such as brain morphology (*Prasad and Keshavan, 2008*; *Brown and Thompson, 2010*; *Nenadic et al., 2012*), dendritic spine morphology (*van Spronsen and Hoogenraad, 2010*; *Penzes et al., 2011*), and other synaptic markers have been sought for neuropathology such as schizophrenia and Alzheimer's disease. *Luebke and Foster (2002)* showed that differences between animals in an acetylcholine receptor subunit might account for differences in susceptibility to noise damage. The inability to predict the outcome of neuronal damage is a serious impediment towards developing effective treatments or prophylactic measures. There is also growing concern regarding inter-individual variability in the efficacy and reliability of brain stimulation (*Kim et al., 2014*; *Lopez-Alonso et al., 2014*).

In contrast, neural circuits in invertebrate systems are composed of relatively small numbers of neurons. In such systems, individual identified neurons often play critical roles and researchers have taken advantage of the simplicity and reproducibility of the connectivity. The number, location, and anatomy of individual neurons are very similar among individuals and their synaptic connections are often drawn by connecting one neuron to the next. However, recent results in several labs have shown that there is considerably more nuance to this view of uniformity than initially appreciated. For example, in the snail respiratory CPG, it was shown there is individual variability at a specific synapse due to distinct activation of two different dopamine receptors (*Magoski and Bulloch, 1999*). In the stomatogastric nervous system of crabs, it was suggested that similar neuronal firing patterns could be obtained from neurons with different compositions of ion channels and that different network configurations could produce similar patterns of activity (*Golowasch et al., 2002*; *Prinz et al., 2004*). More recently, it was shown that the properties of neurons vary substantially even though the overall behavior of the network is constant (*Goaillard et al., 2009*). In the leech heartbeat CPG, it was found that there is considerable variability among individuals in the strength of synaptic connections among interneurons (*Norris et al., 2011*). Despite such animal-to-animal variability, the CPGs produce the same or similar functional outputs. Interestingly, it has been recently shown in the crab stomatogastric nervous system that the bursting activities of circuit neurons showed different susceptibility to extremely high temperature, but such differences were hidden at temperatures within the physiological range (*Tang et al., 2012*). A modeling study further demonstrated that different sets of network parameters could underlie differential sensitivity of neural circuit to extreme temperatures, but are "good enough" to produce robust bursting activities under normal conditions (*Rinberg et al., 2013*).

In the present study, we have shown that in *Tritonia*, the C2 to VSI synapse serves as a hidden phenotype that helps predict the outcome of this particular lesion. Thus, it is possible that a better understanding of the nature of variability in neural circuits could lead to the discovery of a wider range of hidden phenotypes that predict a variety of conditions and allow them to be treated effectively. Variability even in well-defined circuits indicates that there are many solutions to generating a simple function. These solutions might not all be equivalent; under different circumstances, such as injury, one solution might be superior to another.

## What causes synaptic variability?

The causes for the individual synaptic differences in *Tritonia* are not known. It is possible that the synapses differ because of prior experience or that the synapses change autonomously over time. Such fluid neuronal network structure has been suggested in many studies on both mammals and invertebrates. In the mammalian cortex and spinal cord, trial-to-trial variability in the size and/or pattern of the activated neuronal population has been reported (*Shadlen and Newsome, 1998*; *Bair et al., 2001*; *Cai et al., 2006*; *Hansen et al., 2012*). This indicates that continuous changes in functional connectivity within the motor pools and in their activation patterns might underlie such variability (*Cai et al., 2006*; *Cramer, 2008a*). In invertebrates, *Magoski and Bulloch (2000)* showed that specific synapses in the snail respiratory CPG constantly change in sign and that even the valence of synaptic transmission can be modulated by environmental and neurohumoral conditions.

It is equally possible that the differences in synaptic properties between individuals are caused by genetic differences in this wild-caught population. If the differences are genetic, then they could form

the raw material for natural selection to act upon while the behavior remains constant. Such variability, which is neither useful nor injurious, can be hidden in normal life but cause differential survival of individuals when challenged by new conditions (*Darwin, 1876*). Although the experimental procedure of blocking the brain commissure is far from a natural stress to the nervous system, our results shed light on how neural circuits can contain hidden variability, which may lead to differential survival under changing conditions including injury.

## Materials and methods

### Animal surgery

Specimens of the nudibranch, *Tritonia diomedea*, were obtained from Living Elements Ltd. (Delta, BC, Canada) and Friday Harbor Laboratories (San Juan, WA, USA). For behavioral tests of lesions, we employed the same procedure as described previously (*Sakurai and Katz, 2009b*). Animals were placed for 1 hr in ice-chilled artificial seawater (Instant Ocean Salt Water, Miami, USA) containing 0.1% 1-Phynoxy-2-propanol (*Redondo and Murray, 2005*; *Wyeth et al., 2009*). A 2-cm incision was made in the skin between the rhinophores to expose the brain. In the experimental animals, the pedal commissure, which is also known as Pedal Nerve 6 or PdN6 (*Willows et al., 1973*), was cut near the right pedal ganglion with fine scissors, whereas in the sham controls, PdN6 was exposed but not cut. The skin incision was stitched with silk thread and sealed with 'cyanoacrylate glue' (Ethyl Cyanoacrylate, WPI, Sarasota, USA).

Animals with lesions were paired with sham-operated animals for the behavioral assay. The observer was blind to the condition of each animal in a trial. The swim was induced by applying 5 M NaCl solution (0.5 ml) to the dorsal body surface, and the number of swim cycles was counted from complete ventrally directed body flexions. A retraction of the body without a body flexion in response to the stimulus was considered as a swim failure. The number of body flexions during the escape behavior was measured three times before the surgery (16 ± 1, 12 ± 1 and 2 ± 1 hr prior to the commissure transection) and once after the surgery (2 ± 1 hr).

### Isolated brain preparations

Before dissection, the animal was chilled to 4°C in the refrigerator. The brain, consisting of the fused cerebropleural and pedal ganglia, was removed from the chilled animal and immediately pinned to the bottom of a Sylgard-lined chamber (1 ml) where it was superfused with saline at 4°C. Physiological saline composition was (in mM): 420 NaCl, 10 KCl, 10 $CaCl_2$, 50 $MgCl_2$, 10 D-glucose, and 10 HEPES, pH. 7.4. The cell bodies of the neurons were exposed by removing the connective tissue sheath from the surface of the ganglia (*Willows et al., 1973*). The preparation was left >3 hr superfused in saline at 8–10°C before the electrophysiological experiment.

Suction electrodes made from polyethylene tubing were placed on both left and right Pedal Nerves 3 (PdN3). A suction electrode fabricated from a pulled, fire-polished, borosilicate glass tube (i.d., 1.0 mm; o.d., 1.5 mm) was placed in the *en passant* configuration on PdN6. In this study, the other pedal commissure, Pedal Nerve 5 (PdN5, *Willows et al., 1973*) was cut in all isolated brain preparations during dissection. Cutting it after the dissection had no apparent effect on the swim motor pattern (N = 6).

Neurons were identified by soma location, electrophysiological monitoring of axonal projection (cf., *Figure 1C*), coloration, synaptic connectivity, and activity pattern at rest and during the swim motor program as previously described (*Getting, 1981*, *1983b*). There are three types of CPG neurons: Dorsal Swim Interneurons (DSIs, http://www.neuronbank.org/Tri0001043), Cerebral Neuron 2 (C2, http://www.neuronbank.org/Tri0002380), and Ventral Swim Interneuron-B (VSI-B, http://www.neuronbank.org/Tri0002436). For simplicity, we will refer to VSI-B as VSI for this paper. C2 and DSI have cell bodies on the dorsal surface of the cerebral ganglion and project their axons toward the contralateral pedal ganglion, whereas VSI has its cell body on the ventral side of the pleural ganglion and projects its axon toward the ipsilateral pedal ganglion (*Figure 1C*). To record from both C2 and VSI, the brain was twisted around the cerebral commissure as described by *Getting (1983b)*.

The axons of C2, DSI and VSI exit the pedal ganglion through one of two commissural nerves (PdN5 and PdN6) and reach the other pedal ganglion (*Newcomb et al., 2006*; *Hill and Katz, 2007*). To identify neurons, the swim motor program was evoked by stimulating body wall nerve PdN3 with a train of voltage pulses (5–15 V, 1.5 msec) at 5 Hz for 3 s via a suction electrode. Unilateral electrical stimulation of PdN3 is sufficient to elicit the bilaterally symmetric swim motor pattern (*Figure 1B*). Electrical stimuli were given at intervals of greater than 10 min to avoid habituation of the swim motor pattern (*Frost et al., 1996*).

In the isolated brain preparation, PdN6 was functionally disconnected by either physical transection or by blocking impulse propagation with TTX ($1 \times 10^{-4}$M) in a suction pipette that contained the commissure. There was no statistical difference between cutting and pharmacological disconnection in either the number of bursts or the intra-burst spike frequency (*Sakurai and Katz, 2009b*). In this study, both procedures were referred to as PdN6 disconnection.

When TTX was used, blockade of axonal impulses in PdN6 was confirmed by examining the change in the impulse waveform recorded on the commissure (*Figure 3B*, insets). Under control conditions, the impulses were triphasic, with two downward phases (arrowheads) and a positive deflection between them (*Figure 3B*, PdN6 intact). When the action potentials were blocked inside the pipette, the impulses became biphasic with a downward deflection followed by an upward deflection, suggesting that the impulses came into the pipette and were blocked inside it (*Figure 3B*, PdN6 blocked; see *Sakurai and Katz, 2009b*).

In some experiments, the bathing medium was switched to saline containing a high concentration of divalent cations (Hi-Di saline), which raises the threshold for spiking and reduces spontaneous neural firing. The composition of the Hi-Di saline was (in mM): 285 NaCl, 10 KCl, 25 $CaCl_2$, 125 $MgCl_2$, 10 D-glucose, and 10 HEPES (pH 7.4) (*Sakurai and Katz, 2003*). For all experiments the ganglia were superfused at 2 ml/min at 10°C.

## Electrophysiological recordings and stimulations

Neurons were impaled with glass microelectrodes filled with 3 M potassium chloride (12–44 MΩ). To test C2-evoked synapses, a standard stimulation was used; C2 made to fire at 10 Hz using repeated injection of 20 ms current pulses. Axoclamp-2B amplifiers (Molecular Devices, Sunnyvale, USA) were used for all electrophysiological experiments. Recordings were digitized at 2–6 kHz with a 1401plus A/D converter from Cambridge Electronic Design (CED, Cambridge, UK). Data acquisition and analysis were performed with Spike2 software (CED) and SigmaPlot (Jandel Scientific, San Rafael, CA).

A cluster of two or more action potentials with intervals of less than 1 s was considered as a burst. VSI often exhibited a few spikes during nerve stimulation; they were not counted as a burst.

To measure the polysynaptic action of C2 onto VSI, the amplitude and the frequency of spontaneous EPSPs in VSI were measured for 10 s after the onset of the stimulation. Care was taken not to include stimulus artifacts. EPSPs smaller than 0.1 mV were excluded from the analysis. No polysynaptic IPSPs were seen in VSI when C2 was stimulated. We did not distinguish between electrical or chemical synapses in this study. To measure the direct synaptic action of C2 onto VSI, Hi-Di saline was used to remove polysynaptic input. The amplitude of depolarization was measured from the basal resting potential to the maximal peak whereas the amplitude of hyperpolarization was measured from peak to trough.

## Injection of artificial synaptic currents

Dynamic clamp software StdpC (*Kemenes et al., 2011*) was used to mimic the inhibitory component of the direct synapse of C2 onto VSI. The conductance was based on the *Tritonia* swim CPG model designed by *Getting (1989c)* and modified by *Calin-Jageman et al. (2007)*. The current injected into the postsynaptic VSI, $I_{syn}$, is calculated in each dynamic clamp cycle using a first order kinetics model of the release of neurotransmitter (*Destexhe et al., 1994*; *Sharp et al., 1996*; *Kemenes et al., 2011*):

$$I_{syn} = g_{syn}\ S(t)[V_{syn} - V_{post}(t)], \tag{1}$$

where $S(t)$ is the instantaneous synaptic activation, $g_{syn}$ is the maximum synaptic conductance, $V_{syn}$ is the reversal potential (−80 mV) of the synapse, and $V_{post}$ was fixed to the resting membrane potential of VSI. When subtracting a conductance, a negative value of the same magnitude was used for $g_{syn}$.

The instantaneous activation, $S(t)$ is given by the differential equation:

$$(1 - S_\infty(V_{pre}))\tau_{syn}\frac{dS}{dt} = (S_\infty(V_{pre}) - S(t)), \tag{2}$$

where,

$$S_\infty\left(V_{pre}\right) = \begin{cases} \tanh\left[\dfrac{V_{pre} - V_{thresh}}{V_{slope}}\right] & if\ V_{pre} > V_{Thresh} \\ 0 & otherwise \end{cases} \tag{3}$$

$S_\infty$ is the steady state synaptic activation and $\tau_{syn}$ is the time constant for synaptic decay. We included two inhibitory components with different $\tau_{syn}$ values, 700 ms for the faster component and 1300 ms for the slow component. $V_{pre}$ is the presynaptic membrane potential and $V_{thresh}$ is the threshold potential for the release of neurotransmitter; it was set to the level of 50% height of the smallest C2 action potentials. $V_{slope}$ is the synaptic slope parameter of the activation curve. $g_{syn}$ (1000 ± 100 nS) and $V_{slope}$ (20 ± 5 mV) were adjusted before experiment to match the time course of natural inhibition amplitude seen in VSI in that preparation. All synaptic parameters were set identically in each experiment. Because of the locations of the neurons (C2 is on the dorsal side of the brain and VSI is on the ventral side), we were able to manipulate only one C2/VSI pair at a time.

## Statistics

Statistical comparisons were performed by using SigmaPlot ver. 12 (Jandel Scientific, San Rafael, CA) for Student's $t$-test, linear regression, Levene median test, One-way ANOVA, and One-way or two-way repeated measures ANOVA followed by all pairwise multiple comparison (Holm–Sidak method or Tukey test). In all cases, $p < 0.05$ was considered significant. Results are expressed as the mean ±standard deviation.

# Additional information

### Funding

| Funder | Grant reference number | Author |
|---|---|---|
| National Science Foundation (NSF) | IOS-1120950 | Paul S Katz |
| Brains and Behavior Seed Grant Program, Georgia State University | | Paul S Katz |
| March of Dimes Foundation | 6-FY14-441 | Paul S Katz |

The funder had no role in study design, data collection and interpretation, or the decision to submit the work for publication.

### Author contributions

AS, Conception and design, Acquisition of data, Analysis and interpretation of data, Drafting or revising the article; ANT, Acquisition of data, Analysis and interpretation of data; PSK, Conception and design, Analysis and interpretation of data, Drafting or revising the article

### Author ORCIDs

Akira Sakurai, http://orcid.org/0000-0003-2858-1620

### Ethics

Animal experimentation: The animals are housed in appropriately sized tanks equipped with temperature control system. They are routinely fed and cold-anesthetized prior to dissection.

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
