## [Decision Letter]

Thank you for sending your work entitled “Cryptic differences in a neural circuit underlie differential behavioral susceptibility to a neural injury” for consideration at *eLife.* Your article has been favorably evaluated by a Senior editor and 3 reviewers, one of whom is a member of our Board of Reviewing Editors.

The following individuals responsible for the peer review of your submission have agreed to reveal their identity: Ronald L Calabrese, Reviewing editor; Ansgar Buschges, peer reviewer.

The Reviewing editor and the other reviewers discussed their comments before we reached this decision, and the Reviewing editor has assembled the following comments to help you prepare a revised submission.

The authors describe experiments showing that individual variation in the strength of a key inhibitory synapse in a CPG circuit predicts the response of the individual to a specific injury to the network. Specifically, working with the escape swim CPG of the mollusk Tritonia diomedea they show individual variation in the response of intact animals or isolated nervous systems to cutting/blockade of the commissural nerve PdN6. They then show that the impairment brought on by the injury correlates inversely with the strength of an identified inhibitory synapse in the CPG network. Further they show that Dynamic Clamp addition or subtraction of this synapse can predictably reverse or enhance the response to disconnecting of PdN6.

The paper is highly interesting and innovative and provides a mechanistic explanation to why there are individual variations in the response of the nervous systems to injury. It should be of wide interest to the neuroscience community.

The three reviewers are from somewhat different backgrounds and have different overlapping expertise, and they have reached a consensus view that the paper with proper revision would have considerable significance. While the different reviews reflect the differing backgrounds and expertise for the reviewers, they all agree that four major revisions are needed.

Major revisions:

1) There is concern that the authors have not made a compelling case for the broader impacts of their work. They state that their findings have implications for traumatic nervous system injuries and neurodegenerative diseases but fail to make a strong link. The broader impacts of their work might be enhanced by relating their findings more directly to the burgeoning literature on animal-to-animal variability in the nervous system. A stronger case should also be made for the connection between synaptic variability and impairment caused by lesions in Discussion.

2) The data on the direction of spike propagation in VSI (Figure 5, Figure 5–figure supplement 1 and Figure 6) where presented appear as digressions. Some of this information necessary for the Discussion can come by citing previous work or new results summarized without data. Perhaps limit the data to Figure 6 and place it after current Figure 10 Supplement, where it makes logical sense. One could argue then that preps with strong inhibition (C2 to VSI ipsilateral) should have fewer orthodromic bursts and those with fewer orthodromic burst should be more susceptible to PdN6 disconnection. When presented up front there is no context for the direction of propagation in VSI and one starts to focus on the wrong things; i.e., not on synaptic interactions. Figure 13 is used simply to summarize the results, but it could be used to offer a clearer explanation (or proposed hypothesis) for the role of the direction of propagation, and its transition, in the observed phenomenon.

Related to this issue are observations made in Sakurai A, Katz PS. J Neurosci. 2009 Oct 21;29(42):13115-25. In that paper the authors showed an orderly transition from orthodromic to antidromic from the beginning to the end of the motor pattern. Here this order is not mentioned, only the prevalence of anti vs ortho spikes. Is this order somehow related to the more direct correlation of impairment and inhibitory synaptic strength? If so, how? The Figure 11 legend states that all preparations treated with dynamic clamp inhibition showed antidromic spikes but only a fraction that were not treated. This observation should be explained with reference to Figure 13.

3) The paper is somewhat tedious to read because there is so much data presented. The authors are to be commended for being thorough, but there is a possibility of losing the big picture in the details of the data. One potential solution would be to reduce the number of correlograms. If susceptibility and impairment are well defined and rigorously separated then it might be possible to eliminate all the correlograms where the number of VSI bursts are considered and only present those where impairment is considered. The deleted data could simply be mentioned in the text rather than presented as supplemental data.

4) It would strengthen the paper if the authors formally defined susceptibility and impairment as it related to their experiments and used these terms restrictively; i.e., specifically non-synonymously. There is some concern that the term cryptic is used somewhat misleadingly and could be dropped.

The primary reviews (major concerns) are provided to the authors so that they can gain some insight into the consensus presented above. There are some additional insightful comments in these concerns that the authors should consider but they should focus on the consensus.

Other issues to address:

*Reviewer #1*:

The authors describe experiments showing that individual variation in the strength of a key inhibitory synapse in a CPG circuit predicts the response of the individual to a specific injury to the network. Specifically, working with the escape swim CPG of the mollusk Tritonia diomedea they show individual variation in the response of intact animals or isolated nervous systems to cutting/blockade of the commissural nerve PdN6. They then show that the impairment brought on by the injury correlates inversely with the strength of an identified inhibitory synapse in the CPG network. Further they show that Dynamic Clamp addition or subtraction of this synapse can predictably reverse or enhance the response to disconnecting of PdN6.

The paper is nicely illustrated and all necessary data are presented; in fact one might argue that too much data is presented. The writing is concise and clear, perhaps too concise, and the Discussion (the strongest part of the manuscript in terms of writing) is very illuminating. I encourage the authors to delete some of the extraneous data on spike direction in VSI and to add more explanation of the other data. There are scores of correlograms presented and oodles of statistics that make the text tedious without more rationale and explanation.

The paper is highly interesting and innovative and provides a mechanistic explanation to why there are individual variations in the response of the nervous systems to injury. It should be of wide interest to the neuroscience community.

Major revisions:

The material on the direction of spike propagation in VSI (Figure 5, Figure 5–figure supplement 1, and Figure 6) seem unnecessary digressions in a data packed paper. Most of this stuff necessary for the Discussion can come by citing previous work or new results summarized without data. Perhaps a compromise is to limit the data to Figure 6 and to place it after current Figure 10 supplement, where it makes logical sense. One could argue then that preps with strong inhibition (C2 to VSI ipsilateral) should have fewer orthodromic bursts and those with fewer orthodromic burst should be more susceptible to PdN6 disconnection. When presented up front there is no context for the direction of propagation in VSI and one starts to focus on the wrong things; i.e., not on synaptic interactions.

*Reviewer #2*:

The study by Sakuri and coworkers reports a novel finding: the sensitivity of a neural network to injury depends on the relative weight of opposing synaptic inputs between two component neurons in spatially separated areas. The authors analyzed the influence of axon transection on the generation of rhythmic activity in a central pattern generating network for a simple mollusk motor behavior. Importantly, this network is one of the few examples across animal kingdom for which sufficient information exists about network topology with respect to single neurons involved. The authors analyze the excitatory and inhibitory synaptic actions of C2 onto VSI neurons in the swim network of the nudibranch Tritonia before and after the lesion of their axons connecting two separated areas of synaptic interaction. From the results it becomes clear that it is the relative strength of synaptic inhibition from C2 to VSI in one of the two areas that determines the sensitivity of the whole circuit towards inactivating the second area of interaction. To test this hypothesis dynamic clamp technique is employed to introduce artificial synaptic interactions between the C2 and the VSI. The results favor the hypothesis. The study provides evidence that contrary to the robust and predictable performance of a neural network in an organism there exists substantial inter-individual variability in a task specific tuning of topological and cellular properties. The study has been conducted in a thorough way, incl. data evaluation and statistics. I have the following issues that need the attention of the authors in order to clarify their findings:

1) I read the reasoning of the authors that their results may be also interesting in the light of searching for causal origins in traumatic nervous system injuries (title and P2) and neurogenerative diseases (P13, bottom). In its present form this link appears indirect to me: How similar are network damages between stroke patients on the microlevel compared to the system studied? Which kind of traumatic brain injury would be comparable to axon lesions?

2) The authors report data on the analysis of the synaptic influences of the C2 on the VSI neuron. The presented data raise some questions. In Figures 8 and 9 C2 stimulation of some seconds induces very long lasting excitation in VSI neurons. Which neurons are generating the late excitation? From the scheme in Figure 1 the appearance of EPSPs in VSI neurons independent from activity in C2 is difficult to be explained. Are there other neurons are intercalated between C2 and VSI? Most of the long lasting late depolarization is not related to C2-activity. This prompts the question what these data contribute to the question in focus, i.e. how the weighting between spatially separated excitatory and inhibitory interactions between the C2 and the VSI neuron may contribute to the observed phenomenon. This is in particular relevant, because experiments were conducted, in which and polysynaptic interactions were excluded by means of high-divalent cation saline.

3) The authors analyze the amplitude of the EPSPs (e.g. Figure9). Given that the amplitude of EPSPs depends on membrane potential, how was taken care of a control for that?

4) In Figure 9 the amplitude of the EPSPs is increasing with stronger depolarization, which raises the question of the underlying mechanism.

*Reviewer #3*:

This is a well written manuscript that describes an interesting phenomenon: the apparent dominance of one (contralateral) region of the dendritic arbor of a neuron over a different (ipsilateral) region in its contribution to the generation of swimming behavior. The second, non-dominant (cryptic as the authors prefer to call it) becomes influential when the first one is removed. One interesting aspect is the large degree of variability observed in the responses to stimulation after the connective (PdN6) is severed or blocked and how that correlates with the inhibitory synaptic strength of the ipsilateral side. Their use of dynamic clamp to demonstrate this relationship is elegant. Overall the work is well done and the figures illustrate their points well. My main concern is with the overall conclusion, namely that the hidden synapse has a special (cryptic) value that is only revealed after disconnecting the two sides. While this may appear to be so, the explanation could more simply be that there is a dominant spike initiation zone in the VSI neuron (the contralateral one). When that zone is eliminated, the other spike initiation zone takes over and it is very variable. Nevertheless, when the nerve is disconnected a large number of excitatory inputs that normally drive the bursting activity are lost and thus the conditions are not comparable. The inhibitory synapse revealed after nerve blockade may be modulated and thus serve a significant role under normal (connected) conditions, which are not elicited under the present experimental conditions. Additionally, the section 'Variation in neural circuits' in the discussion attempts to place the results of this work in the context of animal-to-animal variability recently described in a number of preparations. Given that the concept of animal-to-animal variability refers to the case where various combinations of many (non-linear) components give rise to stable activity, I am not sure I see the connection of that work with the authors'. It seems to me that the present results show a clear relationship between the strength of an inhibitory synapse on VSI neurons with the magnitude of the change in swimming activity observed after nerve disconnection. This effect is simply masked by the stronger (contralateral) synapse.

It is also important to note that there are a number of examples in the literature of neurons that have multiple action potential initiation sites, which can serve different functions, as reported here. Perhaps a discussion in that context would be useful (e.g.[23]. European Journal of Neuroscience, Vol. 30, pp. 808-822; [102]. Nature, 381(6580):322-5; etc).

Finally, the observation that preponderance of orthodromic spike propagation amidst the possible combination of orthodromic and antidromic spike propagation is predictive of susceptibility to the motor pattern to lesion is confusing to me. In their previous work the authors showed an orderly transition from orthodromic to antidromic from the beginning to the end of the motor pattern. Here this order is not mentioned, only the prevalence. Is this order somehow related to the more direct correlation of susceptibility and inhibitory synaptic strength? If so, how? Figure 13 simply summarizes the results. I was expecting that the diagram would offer a clear explanation (or proposed hypothesis) for the role of the direction of propagation, and its transition, in the observed phenomenon.

---

## [Author Response]

*1) There is concern that the authors have not made a compelling case for the broader impacts of their work. They state that their findings have implications for traumatic nervous system injuries and neurodegenerative diseases but fail to make a strong link. The broader impacts of their work might be enhanced by relating their findings more directly to the burgeoning literature on animal-to-animal variability in the nervous system. A stronger case should also be made for the connection between synaptic variability and impairment caused by lesions in Discussion*.

In the Introduction and Discussion, we have expanded our discussion with regard to white matter injury. Such injury causes loss of higher level functions such as cognition and language abilities, which are distributed over multiple areas in the brain by disrupting functional connections of the network. This is somewhat equivalent to the lesion we are studying here because we specifically blocked axonal spike propagation through a specific commissure that also connects disparate brain areas. A difference is that this lesion is very specific and the neurons are identified so that we can directly study the effect on behavior.

*2) The data on the direction of spike propagation in VSI (*Figure 5*,*
Figure 5*–figure supplement 1 and*
Figure 6*) where presented appear as digressions. Some of this information necessary for the Discussion can come by citing previous work or new results summarized without data. Perhaps limit the data to*
Figure 6
*and place it after current*
Figure 10
*Supplement, where it makes logical sense. One could argue then that preps with strong inhibition (C2 to VSI ipsilateral) should have fewer orthodromic bursts and those with fewer orthodromic burst should be more susceptible to PdN6 disconnection. When presented up front there is no context for the direction of propagation in VSI and one starts to focus on the wrong things; i.e., not on synaptic interactions. Figure 13 is used simply to summarize the results, but it could be used to offer a clearer explanation (or proposed hypothesis) for the role of the direction of propagation, and its transition, in the observed phenomenon*.

We have moved the figure of variable spike propagation to later in the paper and removed plots with non-significant linear regression.

*Related to this issue are observations made in Sakurai A, Katz PS. J Neurosci. 2009 Oct 21;29(42):13115-25. In that paper the authors showed an orderly transition from orthodromic to antidromic from the beginning to the end of the motor pattern. Here this order is not mentioned, only the prevalence of anti vs ortho spikes. Is this order somehow related to the more direct correlation of impairment and inhibitory synaptic strength? If so, how? The Figure 11 legend states that all preparations treated with dynamic clamp inhibition showed antidromic spikes but only a fraction that were not treated. This observation should be explained with reference to Figure 13*.

We thank the reviewers for pointing out about the temporal order of the direction of spike propagation. We added a supplementary figure (Figure 8–figure supplement 1) that shows how the percentage of antidromic (or orthodromic) spiking changed during the swim episode and how likely they were eliminated after PdN6 block. There were more bursts with antidromic spikes later in the swim motor pattern and they were more likely to be eliminated by lesion than orthodromic bursts.

*3) The paper is somewhat tedious to read because there is so much data presented. The authors are to be commended for being thorough, but there is a possibility of losing the big picture in the details of the data. One potential solution would be to reduce the number of correlograms. If susceptibility and impairment are well defined and rigorously separated then it might be possible to eliminate all the correlograms where the number of VSI bursts are considered and only present those where impairment is considered. The deleted data could simply be mentioned in the text rather than presented as supplemental data*.

We substantially reduced the number of plots, especially those with no significant linear correlation. We also changed the order of figures for *ex vivo* experiments; now results from intact brains come first followed by results from brains with the lesion.

*4) It would strengthen the paper if the authors formally defined susceptibility and impairment as it related to their experiments and used these terms restrictively; i.e., specifically non-synonymously. There is some concern that the term cryptic is used somewhat misleadingly and could be dropped*.

We have included these definitions in the beginning of the Results section:“In this study we use the term “impairment” to mean a decrease in the number of body flexions swim episode or in the number of VSI bursts per swim motor pattern and the term “susceptibility” for the likelihood of being impaired upon lesion or blockade of a commissure.” We reconsidered the word cryptic and changed it to “hidden”.

*The primary reviews (major concerns) are provided to the authors so that they can gain some insight into the consensus presented above. There are some additional insightful comments in these concerns that the authors should consider but they should focus on the consensus*.

Other issues to address:

Reviewer #1:

*The authors describe experiments showing that individual variation in the strength of a key inhibitory synapse in a CPG circuit predicts the response of the individual to a specific injury to the network. Specifically, working with the escape swim CPG of the mollusk Tritonia diomedea they show individual variation in the response of intact animals or isolated nervous systems to cutting/blockade of the commissural nerve PdN6. They then show that the impairment brought on by the injury correlates inversely with the strength of an identified inhibitory synapse in the CPG network. Further they show that Dynamic Clamp addition or subtraction of this synapse can predictably reverse or enhance the response to disconnecting of PdN6*.

*The paper is nicely illustrated and all necessary data are presented; in fact one might argue that too much data is presented. The writing is concise and clear, perhaps too concise, and the Discussion (the strongest part of the manuscript in terms of writing) is very illuminating. I encourage the authors to delete some of the extraneous data on spike direction in VSI and to add more explanation of the other data. There are scores of correlograms presented and oodles of statistics that make the text tedious without more rationale and explanation*.

*The paper is highly interesting and innovative and provides a mechanistic explanation to why there are individual variations in the response of the nervous systems to injury. It should be of wide interest to the neuroscience community*.

We thank the reviewer for this praise. We have substantially revised the manuscript along the lines that the reviewer suggested. We have removed many of the correlograms and moved much of the statistical analysis to the figure legends where it won’t clog up the text.

Major revisions:

*The material on the direction of spike propagation in VSI (*Figure 5*,*
Figure 5*–figure supplement 1 and*
Figure 6*) seem unnecessary digressions in a data packed paper. Most of this stuff necessary for the Discussion can come by citing previous work or new results summarized without data. Perhaps a compromise is to limit the data to*
Figure 6
*and to place it after current*
Figure 10
*supplement, where it makes logical sense. One could argue then that preps with strong inhibition (C2 to VSI ipsilateral) should have fewer orthodromic bursts and those with fewer orthodromic burst should be more susceptible to PdN6 disconnection. When presented up front there is no context for the direction of propagation in VSI and one starts to focus on the wrong things; i.e., not on synaptic interactions*.

We have moved the spike propagation figure to later in the paper where it makes more logical sense. We have also revised the representation of the data.

Reviewer #2*:*

1) I read the reasoning of the authors that their results may be also interesting in the light of searching for causal origins in traumatic nervous system injuries (title and P2) and neurogenerative diseases (P13, bottom). In its present form this link appears indirect to me: How similar are network damages between stroke patients on the microlevel compared to the system studied? Which kind of traumatic brain injury would be comparable to axon lesions?

We expanded our discussion of the types of lesions in mammalian systems that would be equivalent to the lesion we are studying here. It turns out that in stroke and trauma, much of the damage is to fibers of passage, which connect disparate brain areas. In this system, we are specifically severing axons of neurons that also connect disparate brain areas. An important difference is that this lesion is very specific and the neurons are identified so that we can directly study the effect on behavior.

*2) The authors report data on the analysis of the synaptic influences of the C2 on the VSI neuron. The presented data raise some questions. In*
Figures 8 and 9
*C2 stimulation of some seconds induces very long lasting excitation in VSI neurons. Which neurons are generating the late excitation? From the scheme in*
Figure 1
*the appearance of EPSPs in VSI neurons independent from activity in C2 is difficult to be explained. Are there other neurons are intercalated between C2 and VSI? Most of the long lasting late depolarization is not related to C2-activity. This prompts the question what these data contribute to the question in focus, i.e. how the weighting between spatially separated excitatory and inhibitory interactions between the C2 and the VSI neuron may contribute to the observed phenomenon. This is in particular relevant, because experiments were conducted, in which and polysynaptic interactions were excluded by means of high-divalent cation saline*.

We have attempted to explain these recruited EPSPs better. In Figure 6–figure–supplement 1, we draw a simple circuit diagram explain how intercalated neurons could explain their appearance. The reason that we discuss these data in this study is that at first glance, this polysynaptic excitation might account for the individual variability. Instead, it turns out that it is the inhibition of VSI by C2 that prevents this recruited excitation from causing proximal excitation of VSI that is the most important.

3) The authors analyze the amplitude of the EPSPs (e.g. Figure9). Given that the amplitude of EPSPs depends on membrane potential, how was taken care of a control for that?

We removed this figure as being extraneous.

*4) In*
Figure 9
*the amplitude of the EPSPs is increasing with stronger depolarization, which raises the question of the underlying mechanism*.

It is likely that the recruited EPSPs come from a number of neurons, which could each have different amplitudes. Therefore, it would be difficult to assess the reason why particular EPSPs were large or small. We have removed mention of the recruited EPSP amplitudes from this manuscript.

Reviewer #3:

*This is a well written manuscript that describes an interesting phenomenon: the apparent dominance of one (contralateral) region of the dendritic arbor of a neuron over a different (ipsilateral) region in its contribution to the generation of swimming behavior. The second, non-dominant (cryptic as the authors prefer to call it) becomes influential when the first one is removed. One interesting aspect is the large degree of variability observed in the responses to stimulation after the connective (PdN6) is severed or blocked and how that correlates with the inhibitory synaptic strength of the ipsilateral side. Their use of dynamic clamp to demonstrate this relationship is elegant. Overall the work is well done and the figures illustrate their points well. My main concern is with the overall conclusion, namely that the hidden synapse has a special (cryptic) value that is only revealed after disconnecting the two sides. While this may appear to be so, the explanation could more simply be that there is a dominant spike initiation zone in the VSI neuron (the contralateral one). When that zone is eliminated, the other spike initiation zone takes over and it is very variable. Nevertheless, when the nerve is disconnected a large number of excitatory inputs that normally drive the bursting activity are lost and thus the conditions are not comparable*.

We largely agree with the reviewer. It is simply the case that animals differ in the extent to which the proximal synapse is inhibitory. If it more inhibitory, it prevents the local recruitment of excitatory inputs onto VSI by C2. This makes the swimming behavior of this animal more vulnerable to disruption than one in which there is a fair degree of proximal excitation. The results of these subtle differences in synaptic strength are to cause the dominant spike initiation zone to differ.

*The inhibitory synapse revealed after nerve blockade may be modulated and thus serve a significant role under normal (connected) conditions, which are not elicited under the present experimental conditions*.

We have no evidence of this speculation.

*Additionally, the section 'Variation in neural circuits' in the discussion attempts to place the results of this work in the context of animal-to-animal variability recently described in a number of preparations. Given that the concept of animal-to-animal variability refers to the case where various combinations of many (non-linear) components give rise to stable activity, I am not sure I see the connection of that work with the authors'. It seems to me that the present results show a clear relationship between the strength of an inhibitory synapse on VSI neurons with the magnitude of the change in swimming activity observed after nerve disconnection. This effect is simply masked by the stronger (contralateral) synapse*.

In this paper, we show that there are differences across individuals in the extent of inhibition at a particular synapse and that this affects the extent of recruited polysynaptic excitation, which in turn affects the site of action potential initiation and subsequent direction of propagation. There is no effect of these circuit differences on the behavior of the animal under normal conditions. Thus, in this sense, the results correspond to previous work showing that various combinations of components give rise to stable activity.

Here we go one step further and ask what happens if the system is challenged with an injury? Will the different configurations respond equally? We found that they did not.

*It is also important to note that there are a number of examples in the literature of neurons that have multiple action potential initiation sites, which can serve different functions, as reported here. Perhaps a discussion in that context would be useful (e.g.*[23]*. European Journal of Neuroscience, Vol. 30, pp. 808-822;*
[102]*. Nature, 381(6580):322-5; etc)*.

We increased our discussion of multiple spike initiation zones.

*Finally, the observation that preponderance of orthodromic spike propagation amidst the possible combination of orthodromic and antidromic spike propagation is predictive of susceptibility to the motor pattern to lesion is confusing to me. In their previous work the authors showed an orderly transition from orthodromic to antidromic from the beginning to the end of the motor pattern. Here this order is not mentioned, only the prevalence. Is this order somehow related to the more direct correlation of susceptibility and inhibitory synaptic strength? If so, how? Figure 13 simply summarizes the results. I was expecting that the diagram would offer a clear explanation (or proposed hypothesis) for the role of the direction of propagation, and its transition, in the observed phenomenon*.

We re-examined our data and found that bursts were mostly likely to fail if they were purely antidromic prior to blocking the commissure. The data representation was cleaned up in the new Figure 8 and Figure 8, figure–supplement 1.